# Multiclass Transductive Online Learning

**Steve Hanneke**
Department of Computer Science
Purdue University
West Lafayette, IN 47907
steve.hanneke@gmail.com

**Vinod Raman**
Department of Statistics
University of Michigan
Ann Arbor, MI 48104
vkraman@umich.edu

**Amirreza Shaeri**
Department of Computer Science
Purdue University
West Lafayette, IN 47907
amirreza.shaeiri@gmail.com

**Unqiue Subedi**
Department of Statistics
University of Michigan
Ann Arbor, MI 48104
subedi@umich.edu

## Abstract

We consider the problem of multiclass transductive online learning when the number of labels can be unbounded. Previous works by Ben-David et al. [1997] and Hanneke et al. [2023b] only consider the case of binary and finite label spaces, respectively. The latter work determined that their techniques fail to extend to the case of unbounded label spaces, and they pose the question of characterizing the optimal mistake bound for unbounded label spaces. We answer this question by showing that a new dimension, termed the Level-constrained Littlestone dimension, characterizes online learnability in this setting. Along the way, we show that the trichotomy of possible minimax rates of the expected number of mistakes established by Hanneke et al. [2023b] for finite label spaces in the realizable setting continues to hold even when the label space is unbounded. In particular, if the learner plays for $T \in \mathbb{N}$ rounds, its minimax expected number of mistakes can only grow like $\Theta(T)$, $\Theta(\log T)$, or $\Theta(1)$. To prove this result, we give another combinatorial dimension, termed the Level-constrained Branching dimension, and show that its finiteness characterizes constant minimax expected mistake-bounds. The trichotomy is then determined by a combination of the Level-constrained Littlestone and Branching dimensions. Quantitatively, our upper bounds improve upon existing multiclass upper bounds in Hanneke et al. [2023b] by removing the dependence on the label set size. In doing so, we explicitly construct learning algorithms that can handle extremely large or unbounded label spaces. A key and novel component of our algorithm is a new notion of shattering that exploits the sequential nature of transductive online learning. Finally, we complete our results by proving expected regret bounds in the agnostic setting, extending the result of Hanneke et al. [2023b].

## 1 Introduction

Imagine you are a famous musician who has released $K \in \mathbb{N}$ songs. You are now on tour visiting $T \in \mathbb{N}$ cities worldwide based on the pre-specified plan, each with unique musical preferences that you have some understanding of. At each city, you can perform only one song in your concert, and following each performance, the audience provides feedback indicating their preferred song from your repertoire. Your goal is to select the song that aligns with the majority's taste in each

city to maximize satisfaction. How can you effectively select songs to ensure the highest audience satisfaction across most cities having minimal assumptions?

The above example and similar real-world situations, where entities operate according to a possibly adversarially chosen pre-specified schedule, can be formulated in a framework called *Multiclass Transductive Online Learning*. Formally, in this setting, an adversary plays a repeated game against the learner over some $T \in \mathbb{N}$ rounds. Before the game begins, the adversary selects a sequence of $T$ instances $(x_1, \ldots, x_T) \in \mathcal{X}^T$ from some non-empty instance space $\mathcal{X}$ (e.g. images) and reveals it to the learner. Subsequently, during each round $t \in \{1, \ldots, T\}$, the learner predicts a label $\hat{y}_t \in \mathcal{Y}$ from some non-empty label space $\mathcal{Y}$ (e.g. categories of images), the adversary reveals the true label $y_t \in \mathcal{Y}$, and the learner suffers the 0-1 loss, namely $\mathbb{1}\{\hat{y}_t \neq y_t\}$. Importantly, the label space $\mathcal{Y}$ is not required even to be countable; we assume only standard measure theoretic properties for it. Following the well-established frameworks in learning theory, given a concept class $\mathcal{C} \subseteq \mathcal{Y}^{\mathcal{X}}$ of functions $c : \mathcal{X} \to \mathcal{Y}$, the goal of the learner is to minimize the number of mistakes relative to the best-fixed concept in $\mathcal{C}$. If there exists $c \in \mathcal{C}$ such that $c(x_t) = y_t$ for all $t \in \{1, \ldots, T\}$, we say we are in the realizable setting, and otherwise in the agnostic setting. We briefly note that if the learner's predictions are randomized, we focus on the expected value of the mentioned objective.

In this paper, our main contribution is algorithmically answering the following question in the multiclass transductive online learning framework:

*Given a concept class $\mathcal{C} \subseteq \mathcal{Y}^{\mathcal{X}}$, what is the minimum expected number of mistakes achievable by a learner against any realizable adversary?*

For the special case of binary classification ($|\mathcal{Y}| = 2$), this question was first considered by Ben-David et al. [1997] and then later fully resolved by Hanneke et al. [2023b]. Additionally, Hanneke et al. [2023b] considered the case where $|\mathcal{Y}| > 2$, but did not resolve this question when $\mathcal{Y}$ is unbounded. In fact, the bounds by Hanneke et al. [2023b] break down even when $|\mathcal{Y}| \geq 2^T$. As a result, Hanneke et al. [2023b] posed the characterization of the minimax expected number of mistakes in the multiclass setting with an infinite label set as an open question, which we resolve in this paper.

## 1.1 Online Learning and Multiclass Classification

In this work, we study *transductive* online learning framework, where the adversary reveals the entire sequence of instances $(x_1, \ldots, x_T)$ to the learner before the game begins. In the *traditional* online learning framework, the sequence of instances $(x_1, \ldots, x_T)$ are revealed to the learner sequentially, one at a time. That is, on round $t \in \{1, \ldots, T\}$, the learner would have only observed $x_1, \ldots, x_t$. The celebrated work of Littlestone [1988] introduced this framework for binary classification and quantified the best achievable number of mistakes against any realizable adversary for a concept class $\mathcal{C} \subseteq \{0, 1\}^{\mathcal{X}}$ in terms of a combinatorial parameter called the Littlestone dimension. Later, the work of Ben-David et al. [2009] showed that the Littlestone dimension of a concept class $\mathcal{C} \subseteq \{0, 1\}^{\mathcal{X}}$ continues to quantify the expected relative mistakes (i.e expected regret) for the mentioned framework in the more general agnostic setting. More recently, Daniely et al. [2012] and Hanneke et al. [2023a] extended these results to multiclass online learning in the realizable and agnostic settings, respectively. See Section A for more details.

In traditional online classification, there are two sources of uncertainty: one associated with the sequence of instances, and the other with respect to the true labels. Ben-David et al. [1997] initiated the study of transductive online classification with the aim of understanding how exclusively label uncertainty impacts the optimal number of mistakes. Furthermore, removing the uncertainty with respect to the instances can significantly reduce the optimal number of mistakes. For example, for the concept class of halfspaces in the realizable setting, the optimal number of mistakes grows linearly with the time horizon $T$ in the traditional online binary classification framework, while only growing as $\Theta(\log T)$ in the transductive online binary classification framework. So, it is natural to reduce the optimal number of mistakes or extend learnable classes whenever we have additional assumptions. Notably, Ben-David et al. [1997] initially called this setting "offline learning", but it was later renamed "Transductive Online Learning" by Hanneke et al. [2023b] due to its close resemblance to *transductive* PAC learning [Vapnik and Chervonenkis, 1974, Vapnik, 1982, 1998]. See Section A for more details.

While Ben-David et al. [1997] and Hanneke et al. [2023b] mainly focused on binary classification, in this work, we focus on the more general multiclass classification setting. Natarajan and Tadepalli

[1988], Natarajan [1989] and Daniely et al. [2012] initiated the study of multiclass prediction within the foundational PAC framework and traditional online framework, respectively. More recently, following the work by [Brukhim et al., 2021], there has been a growing interest in understanding multiclass learning when the size of the label space is unbounded, including Hanneke et al. [2023c,a], Raman et al. [2023]. This interest is driven by several motivations. Firstly, guarantees for the multiclass setting should not inherently depend on the number of labels, even when it is finite. Secondly, in mathematics, concepts involving infinities often provide cleaner insights. Thirdly, insights from this problem might also advance understanding of real-valued regression problems [Attias et al., 2023]. Finally, on a practical front, many crucial machine learning tasks involve classification into extremely large label spaces. For instance, in image object recognition, the number of classes corresponds to the variety of recognizable objects, and in language models, the class count expands with the dictionary size. See Section A for more details.

## 1.2 Main Results and Techniques

In the following subsection, we present an overview of our main findings along with a summary of our proof techniques.

### 1.2.1 Realizable Setting

In the realizable setting, we assume that the sequence of labeled instances $(x_1, y_1), \ldots, (x_T, y_T)$, played by the adversary, is consistent with at least one concept in $\mathcal{C}$. Here, our objective is to minimize the well-known notion of the expected number of mistakes. We provide upper and lower bounds on the best achievable worst-case expected number of mistakes by the learner as a function of $T$ and $\mathcal{C}$, which we denote by $\mathrm{M}^{\star}(T, \mathcal{C})$.

Hanneke et al. [2023b] established a trichotomy of rates in the case of *binary* classification. That is, for every $\mathcal{C} \subseteq \{0,1\}^{\mathcal{X}}$, we have that $\mathrm{M}^{\star}(T, \mathcal{C})$ can only grow like $\Theta(T)$, $\Theta(\log T)$, or $\Theta(1)$; where the Littlestone and Vapnik-Chervonenkis (VC) dimensions of $\mathcal{C}$ characterize the possible rate. In this work, we extend this trichotomy to the multiclass classification setting, even when $\mathcal{Y}$ is unbounded. To do so, we introduce two new combinatorial parameters, termed the Level-constrained Littlestone dimension and Level-constrained Branching dimension.

To define the Level-constrained Littlestone dimension, we first need to define the Level-constrained Littlestone tree. A Level-constrained Littlestone tree is a Littlestone tree with the additional requirement that the same instance has to label all the internal nodes across a given level. Then, the Level-constrained Littlestone dimension is just the largest natural number $d \in \mathbb{N}$, such that there exists a shattered Level-constrained Littlestone tree $T$ of depth $d$. To define the Level-constrained Branching dimension, we first need to define the Level-constrained Branching tree. The Level-constrained Branching tree is a Level-constrained Littlestone tree without the restriction that the labels on the two outgoing edges are distinct. Then, the Level-constrained Branching dimension is then the smallest natural number $d \in \mathbb{N}$ such that for every shattered Level-constrained Branching tree $T$, there exists a path down $T$ such that the number of nodes whose outgoing edges are labeled by different elements of $\mathcal{Y}$ is at most $d$. The Level-constrained Littlestone dimension reduces to the VC dimension when $|\mathcal{Y}| = 2$. Additionally, the finiteness of the Level-constrained Branching and Littlestone dimension coincide when $|\mathcal{Y}| = 2$. Finally, we note that the Level-constrained Branching dimension is exactly equal to the notion of rank in the work of Ben-David et al. [1997]. However, we believe it is simpler to understated. Using the Level-constrained Littlestone and Branching dimension, we establish the following trichotomy.

**Theorem 1.** *(Trichotomy) Let $\mathcal{C} \subseteq \mathcal{Y}^{\mathcal{X}}$ be a concept class. Then, we have:*

$$
\mathrm{M}^{\star}(T, \mathcal{C}) \in \begin{cases} \Theta(1), & \textit{if } \mathrm{B}(\mathcal{C}) < \infty. \\ \Theta(\log T) & \textit{if } \mathrm{D}(\mathcal{C}) < \infty \textit{ and } \mathrm{B}(\mathcal{C}) = \infty. \\ \Theta(T), & \textit{if } \mathrm{D}(\mathcal{C}) = \infty. \end{cases}
$$

*Here, $\mathrm{B}(\mathcal{C})$ is Level-constrained Branching dimension, and $\mathrm{D}(\mathcal{C})$ is Level-constrained Littlestone dimension defined in Section 2.*

To prove the $O(\log T)$ upper bound for binary online classification, Hanneke et al. [2023b] run the Halving algorithm on the projection of $\mathcal{C}$ onto $x_1, ..., x_T$ and use the Sauer–Shelah–Perles (SSP)

lemma to bound the size of this projection by $O(T^{\mathrm{VC}(\mathcal{C})})$. However, this approach is not applicable when $\mathcal{Y}$ is unbounded. For example, when $\mathcal{C} = \{x \mapsto n \,:\, n \in \mathbb{N}\}$ is the set of all constant functions over $\mathbb{N}$, the size of the projection of $\mathcal{C}$ onto even a single $x \in \mathcal{X}$ is infinity. Moreover, the mentioned class can be learned with at most one number of mistakes. Thus, fundamentally new techniques are required. To this end, we define a new notion of shattering which makes it possible to apply an analog of the Halving algorithm. Additionally, while the proof of the $O(1)$ upper bound in Hanneke et al. [2023b] follows immediately from the guarantee of Standard Optimal Algorithm (SOA) by Littlestone [1988], our $O(1)$ upper bound in terms of the Level-constrained Branching dimension requires a modification of the SOA. We complement our results by presenting matching lower bounds. See Section 3 for more details.

In Section F, we provide a comprehensive comparison between our dimensions and existing multiclass combinatorial complexity parameters.

### 1.2.2 Agnostic Setting

In the agnostic setting, we make no assumptions about the sequence $(x_1, y_1), (x_2, y_2), \ldots, (x_T, y_T)$ played by the adversary. Here, our focus shifts to the well-established notion of expected regret, which compares the expected number of mistakes made by the algorithm to that made by the best concept in the concept class over the sequence. As in the realizable setting, we aim to establish both upper and lower bounds on the optimal worst-case expected regret achievable by the learner, expressed as a function of $T$ and the concept class $\mathcal{C}$, denoted by $\mathrm{R}^\star(T, \mathcal{C})$.

The prior work by Hanneke et al. [2023b] showed that in the case of binary classification, $\mathrm{R}^\star(T, \mathcal{C})$ is $\tilde{\Theta}(\sqrt{\mathrm{VC}(\mathcal{C})\,T})$ whenever $\mathrm{VC}(\mathcal{C}) < \infty$ and $\Theta(T)$ otherwise, where $\tilde{\Theta}$ hides logarithmic factors in $T$. Using the Level-constrained Littlestone dimension in hand, we extend these results to multiclass classification.

**Theorem 2.** *For every concept class $\mathcal{C} \subseteq \mathcal{Y}^{\mathcal{X}}$ and $T \geq \mathrm{D}(\mathcal{C})$, we have the following:*

$$\sqrt{\frac{T\,\mathrm{D}(\mathcal{C})}{8}} \leq \mathrm{R}^\star(T, \mathcal{C}) \leq \sqrt{T\,\mathrm{D}(\mathcal{C})}\, \log\Big(\frac{eT}{\mathrm{D}(\mathcal{C})}\Big),$$

*where $\mathrm{D}(\mathcal{C})$ is Level-constrained Littlestone dimension defined in Section 2.*

Our results in the agnostic setting can be proved using core ideas in the proof of the agnostic results from Ben-David et al. [2009], Hanneke et al. [2023a], and Hanneke et al. [2023b]. See Section E for more details.

## 2 Preliminaries

### 2.1 Notation

Let $\mathcal{X}$ denote an example space and $\mathcal{Y}$ denote the label space. We make no assumptions about $\mathcal{Y}$, so it can be unbounded and even uncountable (e.g. $\mathcal{Y} = \mathbb{R}$). Following the work of Hanneke et al. [2023a], if we consider randomized learning algorithms, the associated $\sigma$-algebra is of little consequence, except that singleton sets $\{y\}$ should be measurable. Let $\mathcal{C} \subseteq \mathcal{Y}^{\mathcal{X}}$ denote a concept class. We abbreviate a sequence $z_1, \ldots, z_T$ by $z_{1:T}$. Moreover, we also define $z_{<t} := (z_1, \ldots, z_{t-1})$ and $z_{\leq t} := (z_1, \ldots, z_t)$. Finally for $n \in \mathbb{N}$, we let $[n] := \{1, \ldots, n\}$.

### 2.2 Transductive Online Classification

In the transductive online classification setting, a learner $\mathcal{A}$ plays a repeated game against an adversary over $T$ rounds. Before the game begins, the adversary picks a sequence of labeled instances $(x_1, y_1), \ldots, (x_T, y_T) \in (\mathcal{X} \times \mathcal{Y})^T$ and reveals $x_{1:T}$ to the learner. Then, in each round $t \in [T]$, using $x_{1:T}$ and $y_{1:t-1}$, the learner makes a potentially randomized prediction $\mathcal{A}(x_t) \in \mathcal{Y}$. Finally, the adversary reveals the true label $y_t$, and the learner suffers the loss $\mathbb{1}\{\mathcal{A}(x_t) \neq y_t\}$. Given a concept class $\mathcal{C} \subseteq \mathcal{Y}^{\mathcal{X}}$, the goal of the learner is to output predictions such that its *expected regret*,

$$\mathrm{R}_{\mathcal{A}}(T, \mathcal{C}) := \sup_{(x_1, y_1), \ldots, (x_T, y_T)} \left( \mathbb{E}\left[\sum_{t=1}^{T} \mathbb{1}\{\mathcal{A}(x_t) \neq y_t\}\right] - \inf_{c \in \mathcal{C}} \sum_{t=1}^{T} \mathbb{1}\{c(x_t) \neq y_t\} \right),$$

is small. Moreover, we define $\mathrm{R}^\star(T,\mathcal{C}) := \inf_{\mathcal{A}} \mathrm{R}_{\mathcal{A}}(T,\mathcal{C})$, where the infimum is taken over all transductive online algorithms. We say that a concept class is transductive online learnable in the agnostic setting if $\mathrm{R}^\star(T,\mathcal{C}) = o(T)$.

If the learner is guaranteed to observe a sequence of examples labeled by some concept $c \in \mathcal{C}$, then we say we are in the realizable setting, and the goal of the learner is to minimize its *expected cumulative mistakes*

$$\mathrm{M}_{\mathcal{A}}(T,\mathcal{C}) := \sup_{c \in \mathcal{C}} \sup_{x_{1:T}} \mathbb{E}\left[\sum_{t=1}^{T} \mathbb{1}\{\mathcal{A}(x_t) \neq c(x_t)\}\right].$$

Similarly, we define $\mathrm{M}^\star(T,\mathcal{C}) := \inf_{\mathcal{A}} \mathrm{M}_{\mathcal{A}}(T,\mathcal{C})$, and an analogous definition of transductive online learnability in the realizable setting holds.

## 2.3 Combinatorial Dimensions

Combinatorial dimensions play an important role in providing a tight quantitative characterization of learnability in learning theory. In this section, we review existing combinatorial dimension in online classification and propose two new dimensions that help us establish the minimax rates for transductive online classification. We start by defining the Littlestone dimension which characterizes multiclass online learnability.

**Definition 1** (Littlestone dimension). *The Littlestone dimension of $\mathcal{C}$, denoted $\mathrm{L}(\mathcal{C})$, is the largest $d \in \mathbb{N}$ such that there exists sequences of functions $\{X_t\}_{t=1}^{d}$ where $X_t : \{0,1\}^{t-1} \to \mathcal{X}$ and $\{Y_t\}_{t=1}^{d}$ where $Y_t : \{0,1\}^{t} \to \mathcal{Y}$ such that for every $\sigma \in \{0,1\}^d$, the following holds:*

    *(i)* $Y_t((\sigma_{<t}, 0)) \neq Y_t((\sigma_{<t}, 1))$ *for all $t \in [d]$.*

    *(ii)* $\exists c_\sigma \in \mathcal{C}$ *such that* $c_\sigma(X_t(\sigma_{<t})) = Y_t(\sigma_{\leq t})$ *for all $t \in [d]$.*

*If for every $d \in \mathbb{N}$, there exists sequences $\{X_t\}_{t=1}^{d}$ and $\{Y_t\}_{t=1}^{d}$ satisfying (i) and (ii), we let $\mathrm{L}(\mathcal{C}) = \infty$.*

On the other hand, in this paper, we show that a different dimension, termed the Level-constrained Littlestone dimension, characterizes transductive online classification.

**Definition 2** (Level-constrained Littlestone dimension). *The Level-constrained Littlestone dimension of $\mathcal{C}$, denoted $\mathrm{D}(\mathcal{C})$, is the largest $d \in \mathbb{N}$ such that there exists a sequence of instances $x_1, .., x_d \in \mathcal{X}^d$ and a sequence of functions $\{Y_t\}_{t=1}^{d}$ where $Y_t : \{0,1\}^{t} \to \mathcal{Y}$, such that for every $\sigma \in \{0,1\}^d$, the following holds:*

    *(i)* $Y_t((\sigma_{<t}, 0)) \neq Y_t((\sigma_{<t}, 1))$ *for all $t \in [d]$.*

    *(ii)* $\exists c_\sigma \in \mathcal{C}$ *such that* $c_\sigma(x_t) = Y_t(\sigma_{\leq t})$ *for all $t \in [d]$.*

*If for every $d \in \mathbb{N}$, there exist sequences $\{x_t\}_{t=1}^{d}$ and $\{Y_t\}_{t=1}^{d}$ satisfying (i) and (ii), we let $\mathrm{D}(\mathcal{C}) = \infty$.*

The Littlestone and Level-constrained Littlestone dimensions can also be defined in terms of complete binary trees. A Littlestone tree $\mathcal{T}$ of depth $d$ is a complete binary tree of depth $d$ where the internal nodes are labeled by elements of $\mathcal{X}$ and for every internal node, its two outgoing edges are labeled by distinct elements in $\mathcal{Y}$. Such a tree is *shattered* by $\mathcal{C}$ if for every root-to-leaf path $\sigma \in \{0,1\}^d$, there exists a concept $c_\sigma \in \mathcal{C}$ consistent with the sequence of instance-label pairs obtained by traversing down $\mathcal{T}$ along $\sigma$. The Littlestone dimension is then the largest $d \in \mathbb{N}$ for which there exists a shattered Littlestone tree of depth $d$. From this perspective, the functions $\{X_t\}_{t=1}^{d}$ and $\{Y_t\}_{t=1}^{d}$ in Definition 1 provide the labels on the internal nodes and the outgoing edges of $\mathcal{T}$ respectively. Analogously, a Level-constrained Littlestone tree is simply a Littlestone tree with the additional requirement that the instances labeling the internal nodes are the same across each level. In Definition 2, $x_1$ labels all the internal nodes on level one, $x_2$ labels all the internal nodes on level two, and so forth. The functions $\{Y_t\}_{t=1}^{d}$ provide the labels on the outgoing edges of a Level-constrained Littlestone tree. Then, the Level-constrained Littlestone dimension is the largest $d \in \mathbb{N}$ for which

there exists a shattered Level-constrained Littlestone tree of depth $d$. We will use the function-based and tree-based definitions of these dimensions interchangeably.

Moreover, we show that the Level-constrained Branching dimension characterizes when constant minimax rates are possible in transductive online classification.

**Definition 3** (Level-constrained Branching dimension). *The Level-constrained Branching dimension of $\mathcal{C}$, denoted $\mathrm{B}(\mathcal{C})$, is the smallest $p \in \mathbb{N}$ such that for every $d \in \mathbb{N}$, every sequence of instances $x_1, .., x_d \in \mathcal{X}^d$, and every sequence of functions $\{Y_t\}_{t=1}^d$ where $Y_t : \{0,1\}^t \to \mathcal{Y}$:*

$$\forall \sigma \in \{0,1\}^d, \exists c_\sigma \in \mathcal{C} \text{ such that } c_\sigma(x_t) = Y_t(\sigma_{\leq t}) \text{ for all } t \in [d]$$

$$\implies \underset{\sigma \in \{0,1\}^d}{\arg\min} \sum_{t=1}^d \mathbb{1}\{Y_t((\sigma_{<t}, 0)) \neq Y_t((\sigma_{<t}, 1))\} \leq p.$$

*If no such $p \in \mathbb{N}$ exist, we let $\mathrm{B}(\mathcal{C}) = \infty$.*

For a given path $\sigma \in \{0,1\}^d$, we refer to $\sum_{t=1}^d \mathbb{1}\{Y_t((\sigma_{<t}, 0)) \neq Y_t((\sigma_{<t}, 1))\}$ as the *branching factor* of the path. In terms of trees, a Level-constrained Branching tree is a Level-constrained Littlestone tree without the restriction that the labels on the outgoing edges of any internal node need to be distinct. Given a path in such a tree, the branching factor of a path counts the number of nodes in the path whose two outgoing edges are labeled by distinct labels in $\mathcal{Y}$. Finally, the Level-constrained Branching dimension can be equivalently defined as the smallest $p \in \mathbb{N}$ such that every *shattered* Level-constrained Branching tree $\mathcal{T}$ of depth $d \in \mathbb{N}$ must have at least one path with branching factor at most $p$.

The following proposition, whose proof is in Appendix B, establishes the relationship between the three dimensions.

**Proposition 1.** *For every $\mathcal{C} \subseteq \mathcal{Y}^{\mathcal{X}}$, we have that $\mathrm{D}(\mathcal{C}) \leq \mathrm{B}(\mathcal{C}) \leq \mathrm{L}(\mathcal{C})$.*

We also compare our dimensions to other existing dimensions in multiclass learning in Section F.

## 3 A Trichotomy in the Realizable Setting

We start by establishing upper and lower bounds on the minimax expected number of mistakes in the realizable setting in terms of the Level-constrained Littlestone dimension and the Level-constrained Branching dimension.

**Theorem 3** (Mistake bound). *For every concept class $\mathcal{C} \subseteq \mathcal{Y}^{\mathcal{X}}$, we have*

$$\frac{1}{2} \min \left\{ \max \left\{ \mathrm{D}(\mathcal{C}), \lfloor \log T \rfloor \cdot \mathbb{1}[\mathrm{B}(\mathcal{C}) = \infty] \right\}, T \right\} \leq \mathrm{M}^\star(T, \mathcal{C}) \leq \min \left\{ \mathrm{B}(\mathcal{C}), \mathrm{D}(\mathcal{C}) \log\left(\frac{eT}{\mathrm{D}(\mathcal{C})}\right), T \right\}.$$

One can trivially upper bound $\mathrm{M}^\star(T, \mathcal{C})$ by $\mathrm{L}(\mathcal{C})$. However, by Proposition 1, our upper bound in terms of $\mathrm{B}(\mathcal{C})$ is sharper. We can also infer from the proof in Section 3.3 that when $T$ is large enough (namely $T \gg 2^{\mathrm{B}(\mathcal{C})}$), the lower bound in the realizable setting is also $\frac{\mathrm{B}(\mathcal{C})}{2}$.

Given Theorem 3, we immediately infer a trichotomy in minimax rates.

**Corollary 1** (Trichotomy). *For every concept class $\mathcal{C} \subseteq \mathcal{Y}^{\mathcal{X}}$, we have*

$$\mathrm{M}^\star(T, \mathcal{C}) = \begin{cases} \Theta(1), & \text{if } \mathrm{B}(\mathcal{C}) < \infty. \\ \Theta(\log T) & \text{if } \mathrm{D}(\mathcal{C}) < \infty \text{ and } \mathrm{B}(\mathcal{C}) = \infty. \\ \Theta(T), & \text{if } \mathrm{D}(\mathcal{C}) = \infty. \end{cases}$$

*Proof.* (of Corollary 1) When $\mathrm{B}(\mathcal{C}) < \infty$, Theorem 3 gives that $\frac{1}{2}\mathrm{D}(\mathcal{C}) \leq \mathrm{M}^\star(T, \mathcal{C}) \leq \mathrm{B}(\mathcal{C})$ for $T \geq \mathrm{D}(\mathcal{C})$. When $\mathrm{B}(\mathcal{C}) = \infty$ but $\mathrm{D}(\mathcal{C}) < \infty$, Theorem 3 gives that $\frac{1}{2}\lfloor \log T \rfloor \leq \mathrm{M}^\star(T, \mathcal{C}) \leq \mathrm{D}(\mathcal{C}) \log\left(\frac{eT}{\mathrm{D}(\mathcal{C})}\right)$ for $\lfloor \log T \rfloor \geq \mathrm{D}(\mathcal{C})$. Finally, when $\mathrm{D}(\mathcal{C}) = \infty$, Theorem 3 gives that $\frac{T}{2} \leq \mathrm{M}^\star(T, \mathcal{C}) \leq T$. $\qquad\square$

The remainder of this Section is dedicated to proving Theorem 3. The proof of the lowerbound $\mathrm{D}(\mathcal{C})/2$ follows from standard techniques, so we defer it to Appendix C.

### 3.1 Proof of Upperbound $B(\mathcal{C})$

*Proof.* Fix $n \in \mathbb{N}$, a sequence of instances $x_{1:n} := (x_1, \ldots, x_n) \in \mathcal{X}^n$, a sequence of functions $Y_{1:n} = (Y_1, \ldots, Y_n)$ such that $Y_t : \{0,1\}^n \to \mathcal{Y}$, and set of concepts $V \subseteq \mathcal{C}$. If $\forall \sigma \in \{0,1\}^n$, there exists $c_\sigma \in V$ such that $c_\sigma(x_t) = Y_t(\sigma_{\leq t})$ for all $t \in [n]$, then define $B(V, x_{1:n}, Y_{1:n}) := \arg\min_{\sigma \in \{0,1\}^n} \sum_{t=1}^n \mathbb{1}\{Y_t((\sigma_{<t}, 0)) \neq Y_t((\sigma_{<t}, 1))\}$. Otherwise, define $B(V, x_{1:n}, Y_{1:n}) := 0$. Recall that we can represent $x_{1:n}$ and $Y_{1:n}$ with level-constrained trees $\mathcal{T}$ of depth $n$. With the tree representation, $B(V, x_{1:n}, Y_{1:n}) := 0$ if $V$ does not shatter $\mathcal{T}$. If $\mathcal{T}$ is shattered by $V$, then $B(V, x_{1:n}, Y_{1:n})$ is the minimum branching factor across all the root-to-leaf paths in $\mathcal{T}$. Recall that the branching factor of a path is the number of nodes in this path whose left and right outgoing edges are labeled by two distinct elements of $\mathcal{Y}$. In this proof, we work with an instance-dependent complexity measure of $V \subseteq \mathcal{C}$ defined as

$$B(V, x_{1:n}) := \sup_{Y_{1:n}} B(V, x_{1:n}, Y_{1:n}).$$

We now define a learning algorithm that obtains the claimed mistake bound of $B(\mathcal{C})$. Fix a time horizon $T \in \mathbb{N}$, and let $x_{1:T} = (x_1, x_2, \ldots, x_T)$ denote the sequence of instances revealed by the adversary. Initialize $V_1 := \mathcal{C}$. For every $t \in \{1, \ldots, T\}$, if we have $\{c(x_t) : c \in V_t\} = \{y\}$, then predict $\hat{y}_t = y$. Otherwise, define $V_t^y = \{c \in V_t : c(x_t) = y\}$ for all $y \in \mathcal{Y}$, and predict $\hat{y}_t = \arg\max_{y \in \mathcal{Y}} B(V_t^y, x_{t+1:T})$. Finally, the learner receives a feedback $y_t \in \mathcal{Y}$, and updates $V_{t+1} \leftarrow V_t^{y_t}$. When $t = T$, the sequence $x_{T+1:T}$ is null and we define $\hat{y}_T = \arg\max_{y \in \mathcal{Y}} B(V_T^y)$. For this learning algorithm, we prove that

$$B(V_{t+1}, x_{t+1:T}) \leq B(V_t, x_{t:T}) - \mathbb{1}\{y_t \neq \hat{y}_t\}. \tag{1}$$

Rearranging and summing over $t \in [T]$ rounds, we obtain:

$$\sum_{t=1}^T \mathbb{1}\{y_t \neq \hat{y}_t\} \leq \sum_{t=1}^T \Big( B(V_t, x_{t:T}) - B(V_{t+1}, x_{t+1:T}) \Big) = B(V_1, x_{1:T}) - B(V_{T+1}, x_{T+1:T})$$
$$\leq B(V_1, x_{1:T}) \leq B(\mathcal{C})$$

The equality above follows because the sum telescopes. The final inequality follows because $V_1 = \mathcal{C}$ and the level-constrained branching dimension of $\mathcal{C}$ is defined as $B(\mathcal{C}) = \sup_{T \in \mathbb{N}} \sup_{x_{1:T} \in \mathcal{X}^T} B(\mathcal{C}, x_{1:T})$.

We now prove inequality (1). There are two cases to consider: (a) $y_t = \hat{y}_t$ and (b) $y_t \neq \hat{y}_t$. Starting with (a), let $y_t = \hat{y}_t$. Recall that $B(V_{t+1}, x_{t+1:T}) = B(V_t^{y_t}, x_{t+1:T})$. Since $c(x_t) = y_t$ for all $h \in V_t^{y_t}$, we must have $B(V_t^{y_t}, x_{t+1:T}) \leq B(V_t^{y_t}, x_{t:T})$. Finally, using the fact that $V_t^{y_t} \subseteq V_t$, we have $B(V_t^{y_t}, x_{t:T}) \leq B(V_t, x_{t:T})$. This establishes (1) for this case.

Moving to (b), let $y_t \neq \hat{y}_t$. Note that we must have $B(V_t, x_{t:T}) > 0$. Otherwise, if $B(V_t, x_{t:T}) = 0$, then we have $\{c(x_t) : c \in V_t\} = \{y\}$. So, by our prediction rule, we cannot have $y_t \neq \hat{y}_t$ under realizability. To establish (1), we want to show that $B(V_{t+1}, x_{t+1:T}) < B(V_t, x_{t:T})$. Suppose, for the sake of contradiction, we instead have $B(V_{t+1}, x_{t+1:T}) \geq B(V_t, x_{t:T})$. Then, let us define $d := B(V_{t+1}, x_{t+1:T})$. If $d = 0$, then our proof is complete because $0 \leq B(V_t, x_{t:T}) - \mathbb{1}\{y_t \neq \hat{y}_t\}$. Assume that $d > 0$ and recall that $V_{t+1} = V_t^{y_t}$. By definition of $B(V_t^{y_t}, x_{t+1:T})$ and its equivalent shattered-trees representation, there exists a level-constrained tree $\mathcal{T}_{y_t}$ of depth $T - t$ whose internal nodes are labeled by $x_t, \ldots, x_T$ and is shattered by $V_t^{y_t}$. Moreover, every path down $\mathcal{T}_{y_t}$ has branching factor $\geq d$.

Next, as $\hat{y}_t = \arg\max_{y \in \mathcal{Y}} B(V_t^y, x_{t+1:T})$, we further have $B(V_t^{\hat{y}_t}, x_{t+1:T}) \geq B(V_t^{y_t}, x_{t+1:T}) \geq d$. Thus, there exists another level-constrained tree $\mathcal{T}_{\hat{y}_t}$ of depth $T - t$ whose internal nodes are labeled by $x_t, \ldots, x_T$, that is shattered by $V_t^{\hat{y}_t}$, and every path down $\mathcal{T}_{\hat{y}_t}$ has branching factor $\geq d$. Finally, consider a new tree $\mathcal{T}$ with root-node labeled by $x_t$, the left-outgoing edge from the root node is labeled by $y_t$, and the right outgoing edge is labeled by $\hat{y}_t$. Moreover, the subtree following the outgoing edge labeled by $y_t$ is $\mathcal{T}_{y_t}$, and the subtree following the outgoing edge labeled by $\hat{y}_t$ is $\mathcal{T}_{\hat{y}_t}$. Since both $\mathcal{T}_{y_t}$ and $\mathcal{T}_{\hat{y}_t}$ are valid level-constrained trees each with internal nodes labeled by $x_{t+1}, \ldots, x_T$, the newly constructed tree $\mathcal{T}$ is a also a level-constrained trees of depth $T - t + 1$ with internal nodes labeled by $x_t, \ldots, x_T$. In addition, as $\mathcal{T}_{y_t}$ and $\mathcal{T}_{\hat{y}_t}$ are shattered by $V_t^{y_t}$ and $V_t^{\hat{y}_t}$ respectively, the tree $\mathcal{T}$ must be shattered by $V_t^{y_t} \cup V_t^{\hat{y}_t}$. Finally, as every path down each

sub-trees $\mathcal{T}_{y_t}$ and $\mathcal{T}_{\hat{y}_t}$ has branching factor $\geq d$ and $y_t \neq \hat{y}_t$, every path of $\mathcal{T}$ must have branching factor $\geq d+1$. This shows that $\mathrm{B}(V_t^{y_t} \cup V_t^{\hat{y}_t}, x_{t:T}) \geq d+1$. And since $V_t^{y_t} \cup V_t^{\hat{y}_t} \subseteq V_t$, we have $d+1 \leq \mathrm{B}(V_t^{y_t} \cup V_t^{\hat{y}_t}, x_{t:T}) \leq \mathrm{B}(V_t, x_{t:T})$ by monotonicity. This contradicts our assumption that $d := \mathrm{B}(V_{t+1}, x_{t+1:T}) \geq \mathrm{B}(V_t, x_{t:T})$. Therefore, we must have $\mathrm{B}(V_{t+1}, x_{t+1:T}) < \mathrm{B}(V_t, x_{t:T})$. This establishes (1), completing our proof. $\qquad\square$

## 3.2   Proof of Upperbound $\mathrm{D}(\mathcal{C}) \log\left(\frac{eT}{\mathrm{D}(\mathcal{C})}\right)$

*Proof.* Fix the time horizon $T \in \mathbb{N}$ and let $x_{1:T} := (x_1, ..., x_T) \in \mathcal{X}^T$ be the sequence of $T$ instances revealed to the learner at the beginning of the game. We say a subsequence $x'_{1:n} := (x'_1, ..., x'_n)$, preserving the same order as in $x_{1:T}$, is shattered by $V \subseteq \mathcal{C}$ if there exists a sequence of functions $\{Y_t\}_{t=1}^n$, where $Y_t : \{0,1\}^t \to \mathcal{Y}$, such that for every $\sigma \in \{0,1\}^n$, we have that

(i)  $Y_t(\sigma_{<t}, 0) \neq Y_t(\sigma_{<t}, 1)$ for all $t \in [n]$,

(ii)  $\exists c_\sigma \in V$ such that $c_\sigma(x_t) = Y_t(\sigma_{\leq t})$ for all $t \in [n]$.

For every $V \subseteq \mathcal{C}$, let $\mathrm{S}(V)$ be the number of subsequences of $x_{1:T}$ shattered by $V$. In addition, for every $(x, y) \in \mathcal{X} \times \mathcal{Y}$, let $V_{(x,y)} := \{c \in V : c(x) = y\}$. Consider the following online learner. At the beginning of the game, the learner initializes $V^1 = \mathcal{C}$. Then, in every round $t \in [T]$, the learner predicts $\hat{y}_t \in \arg\max_{y \in \mathcal{Y}} \mathrm{S}(V_{(x_t, y)}^t)$, receives $y_t \in \mathcal{Y}$, and updates $V^{t+1} \leftarrow V_{(x_t, y_t)}^t$.

For this learning algorithm, we claim that

$$\mathrm{S}(V^{t+1}) \leq \max\left\{\mathbb{1}\{y_t = \hat{y}_t\}, \frac{1}{2}\right\} \mathrm{S}(V^t)$$

for every round $t \in [T]$. This implies the stated mistake bound since $\mathrm{S}(\mathcal{C}) \leq \sum_{i=0}^{\mathrm{D}(\mathcal{C})} \binom{T}{i} \leq \left(\frac{eT}{\mathrm{D}(\mathcal{C})}\right)^{\mathrm{D}(\mathcal{C})}$ and the learner can make at most $\log(\mathrm{S}(\mathcal{C}))$ mistakes before $\mathrm{S}(V^t) = 1$. We now prove this claim by considering the case where $\hat{y}_t = y_t$ and $\hat{y}_t \neq y_t$ separately.

Let $t \in [T]$ be a round where $\hat{y}_t = y_t$. Then, $\mathrm{S}(V^{t+1}) \leq \mathrm{S}(V^t)$ since $V^{t+1} = V_{(x_t, y_t)}^t \subseteq V^t$. Now, let $t \in [T]$ be a round where $\hat{y}_t \neq y_t$. We need to show that $\mathrm{S}(V^{t+1}) \leq \frac{1}{2}\mathrm{S}(V^t)$. For any $V \subseteq \mathcal{C}$, let $\mathrm{Sh}(V)$ be the set of all subsequences of $x_{1:T}$ that are shattered by $V$. Then, for any subsequence $q \in \mathrm{Sh}(V^t)$, only one of the following properties must be true:

(1)  $q \notin \mathrm{Sh}(V_{(x_t, y_t)}^t)$ and $q \notin \mathrm{Sh}(V_{(x_t, \hat{y}_t)}^t)$,

(2)  $q \in \mathrm{Sh}(V_{(x_t, y_t)}^t) \Delta \mathrm{Sh}(V_{(x_t, \hat{y}_t)}^t)$,

(3)  $q \in \mathrm{Sh}(V_{(x_t, y_t)}^t) \cap \mathrm{Sh}(V_{(x_t, \hat{y}_t)}^t)$,

where $\Delta$ denotes the symmetric difference. For every $i \in \{1, 2, 3\}$, let $\mathrm{Sh}^i(V^t) \subseteq \mathrm{Sh}(V^t)$ be the subset of $\mathrm{Sh}(V^t)$ that satisfies property $(i)$. Note that $\mathrm{Sh}(V^t) = \bigcup_{i=1}^3 \mathrm{Sh}^i(V^t)$ and $\{\mathrm{Sh}^i(V^t)\}_{i=1}^3$ are pairwise disjoint. Therefore, $\{\mathrm{Sh}^i(V^t)\}_{i=1}^3$ forms a partition of $\mathrm{Sh}(V^t)$. For each $i \in \{1, 2, 3\}$, we compute how many elements of $\mathrm{Sh}^i(V^t)$ we drop when going from $\mathrm{Sh}(V^t)$ to $\mathrm{Sh}(V^{t+1})$. We can then upperbound $|\mathrm{Sh}(V_{(x_t, y_t)}^t)| = \mathrm{S}(V^{t+1})$ by lower bounding $|\mathrm{Sh}(V^t) \setminus \mathrm{Sh}(V^{t+1})|$, the number of elements we drop across all of the subsets $\{\mathrm{Sh}^i(V^t)\}_{i=1}^3$ when going from $V^t$ to $V^{t+1}$.

Starting with $i = 1$, observe that for every $q \in \mathrm{Sh}^1(V^t)$, we have that $q \notin \mathrm{Sh}(V_{(x_t, y_t)}^t)$. Therefore, $|\mathrm{Sh}^1(V^t) \cap \mathrm{Sh}(V_{(x_t, y_t)}^t)| = 0$, implying that we drop all the elements from $\mathrm{Sh}^1(V^t)$ when going from $\mathrm{Sh}(V^t)$ to $\mathrm{Sh}(V^{t+1})$.

For the case where $i = 2$, note that $\mathrm{Sh}(V_{(x_t, y_t)}^t), \mathrm{Sh}(V_{(x_t, \hat{y}_t)}^t) \subseteq \mathrm{Sh}(V^t)$ and $\mathrm{S}(V_{(x_t, \hat{y}_t)}^t) \geq \mathrm{S}(V_{(x_t, y_t)}^t)$, where the latter inequality is true by the definition of the prediction rule. Moreover, using

the fact that $\{\mathrm{Sh}^i(V^t)\}_{i=1}^3$ forms a partition of $\mathrm{Sh}(V^t)$, we can write

$$S(V^t_{(x_t,\hat{y}_t)}) = |\mathrm{Sh}^3(V^t)| + |\mathrm{Sh}^2(V^t) \cap \mathrm{Sh}(V^t_{(x_t,\hat{y}_t)})|$$

and

$$S(V^t_{(x_t,y_t)}) = |\mathrm{Sh}^3(V^t)| + |\mathrm{Sh}^2(V^t) \cap \mathrm{Sh}(V^t_{(x_t,y_t)})|.$$

Since $S(V^t_{(x_t,\hat{y}_t)}) \geq S(V^t_{(x_t,y_t)})$, we get that $|\mathrm{Sh}^2(V^t) \cap \mathrm{Sh}(V^t_{(x_t,\hat{y}_t)})| \geq |\mathrm{Sh}^2(V^t) \cap \mathrm{Sh}(V^t_{(x_t,y_t)})|$. This implies that $|\mathrm{Sh}^2(V^t) \cap \mathrm{Sh}(V^t_{(x_t,y_t)})| \leq \frac{1}{2}|\mathrm{Sh}^2(V^t)|$ since $\left(\mathrm{Sh}^2(V^t) \cap \mathrm{Sh}(V^t_{(x_t,\hat{y}_t)})\right) \cup \left(\mathrm{Sh}^2(V^t) \cap \mathrm{Sh}(V^t_{(x_t,y_t)})\right) = \mathrm{Sh}^2(V^t)$ and $\left(\mathrm{Sh}^2(V^t) \cap \mathrm{Sh}(V^t_{(x_t,\hat{y}_t)})\right) \cap \left(\mathrm{Sh}^2(V^t) \cap \mathrm{Sh}(V^t_{(x_t,y_t)})\right) = \emptyset$. Thus, we drop at least half the elements from $\mathrm{Sh}^2(V^t)$ when going from $\mathrm{Sh}(V^t)$ to $\mathrm{Sh}(V^{t+1})$.

Finally, consider when $i = 3$. Fix a $q \in \mathrm{Sh}^3(V^t)$. We claim that $x_1, ..., x_t \notin q$. This is because every $c \in V^t$ outputs $y_j$ on $x_j$ for all $j \leq t-1$. In addition, $x_t \notin q$ because $q \in \mathrm{Sh}(V^t_{(x_t,y_t)}) \cap \mathrm{Sh}(V^t_{(x_t,\hat{y}_t)})$ and every concept in $V^t_{(x_t,y_t)}$ and $V^t_{(x_t,\hat{y}_t)}$ outputs $y_t$ and $\hat{y}_t$ on $x_t$ respectively. Thus, the sequence $x_t \circ q$, obtained by concatenating $x_t$ to the front of $q$, is a valid subsequence of $x_{1:T}$. Since $\hat{y}_t \neq y_t$, we also have that $x_t \circ q$ is shattered by $V^t$. Using the fact that $x_t \circ q \notin \mathrm{Sh}(V^t_{(x_t,y_t)})$ and $x_t \circ q \notin \mathrm{Sh}(V^t_{(x_t,\hat{y}_t)})$, gives that $x_t \circ q \in \mathrm{Sh}^1(V_t)$. Since our choice of $q$ was arbitrary, this implies that for every $q \in \mathrm{Sh}^3(V_t)$, there exists a subsequence $q' = x_t \circ q \in \mathrm{Sh}^1(V^t)$, ultimately giving that $|\mathrm{Sh}^1(V^t)| \geq |\mathrm{Sh}^3(V^t)|$.

To complete the proof, we lowerbound the total number of dropped elements when going from $\mathrm{Sh}(V^t)$ to $\mathrm{Sh}(V^{t+1})$ by

$$\begin{aligned}
|\mathrm{Sh}(V^t) \setminus \mathrm{Sh}(V^t_{(x_t,y_t)})| &\geq |\mathrm{Sh}^1(V^t)| + \frac{|\mathrm{Sh}^2(V^t)|}{2} \\
&\geq \frac{|\mathrm{Sh}^1(V^t)|}{2} + \frac{|\mathrm{Sh}^2(V^t)|}{2} + \frac{|\mathrm{Sh}^3(V^t)|}{2} \\
&= \frac{|\mathrm{Sh}(V^t)|}{2} = \frac{S(V^t)}{2}.
\end{aligned}$$

The number of remaining elements is then $S(V^{t+1}) = |\mathrm{Sh}(V^t_{(x_t,y_t)})| = |\mathrm{Sh}(V^t)| - |\mathrm{Sh}(V^t) \setminus \mathrm{Sh}(V^t_{(x_t,y_t)})| \leq \frac{1}{2} S(V^t)$, as needed. $\qquad\square$

We end this section by noting that the algorithm in the proof of Theorem 3 can be made conservative (i.e. does not update when it is correct) with the same mistake bound. This conservative-version of the realizable learner will be used when proving regret bounds in the agnostic setting (see Section E).

### 3.3 Proof of Lowerbound $\frac{\lfloor \log T \rfloor}{2} \mathbb{1}[B(\mathcal{C}) = \infty]$

If $B(\mathcal{C}) = \infty$, then for every $q \in \mathbb{N}$, Definition 3 guarantees the existence of $d \in \mathbb{N}$, a sequence of instances $x_1, \ldots, x_d$, and a sequence of functions $Y_1, \ldots, Y_d$ where $Y_t : \{0,1\}^t \to \mathcal{Y}$ such that the following holds: (i) $\forall \sigma \in \{0,1\}^d$, there exists $c_\sigma \in \mathcal{C}$ such that $c_\sigma(x_t) = Y_t(\sigma_{\leq t})$ (ii) $\forall \sigma \in \{0,1\}^d$, we have $\sum_{t=1}^d \mathbb{1}\{Y_t((\sigma_{<t}, 0)) \neq Y_t((\sigma_{<t}, 1))\} \geq q$. Equivalently, there exists a shattered level-constrained branching tree $\mathcal{T}$ of depth $d$ with internal nodes labeled by instances $x_1, \ldots, x_d$ such that every path down the tree $\mathcal{T}$ has $\geq q$ branching factor. Recall that the branching factor of a path is the number of nodes in the path whose two outgoing edges are labeled by two distinct elements of $\mathcal{Y}$. We say that an internal node has branching if the left and right outgoing edges from the node are labeled by two distinct elements of $\mathcal{Y}$.

Without loss of generality, we will assume that the $\mathcal{T}$ guaranteed by Definition 3 has the following properties: (a) every path in $\mathcal{T}$ has exactly $q$ branching factor and (b) $\mathcal{T}$ is symmetric along its non-branching nodes– that is, for every node in $\mathcal{T}$ that has no branching, the subtrees on its left and right outgoing edges are identical. There is no loss in generality because given $\mathcal{T}$ without property (a), we can traverse down every path in $\mathcal{T}$, and once the path has branching factor $q$, label all the

subsequent outgoing edges down the path by a concept in $\mathcal{C}$ that shatters any completion of that path. For property (b), if any non-branching node has two different subtrees, then replace the right subtree with the left subtree. Given such tree $\mathcal{T}$, let $\mathrm{BN}(\mathcal{T})$ denote the number of levels in $\mathcal{T}$ with at least one branching node. The following Lemma, whose proof can be found in Appendix D, provides an upperbound on $\mathrm{BN}(\mathcal{T})$.

**Lemma 1.** *Let $\mathcal{T}$ be any level-constrained branching tree shattered by $\mathcal{C}$ such that: (a) every path in $\mathcal{T}$ has exactly $q \in \mathbb{N}$ branching and (b) for every node in $\mathcal{T}$ without branching, the subtrees on its left and right outgoing edges are identical. Then, $\mathrm{BN}(\mathcal{T}) \leq 2^q - 1$.*

Given this Lemma, we now prove the claimed lowerbound of $\frac{\lfloor \log T \rfloor}{2} \mathbb{1}[\mathrm{B}(\mathcal{C}) = \infty]$. Assume $\mathrm{B}(\mathcal{C}) = \infty$ and take $q = \lfloor \log T \rfloor$. Definition 3 guarantees the existence of shattered Level-constrained branching tree $\mathcal{T}$ that satisfies property (a) and (b) specified in Lemma 1. Next, Lemma 1 implies that $\mathrm{BN}(\mathcal{T}) \leq 2^{\lfloor \log T \rfloor} - 1 \leq T - 1$. Let $d$ be the depth $\mathcal{T}$ and $S \subseteq \{1, \ldots, d\}$ be the levels in $\mathcal{T}$ with at least one branching node. By definition, we have $|S| = \mathrm{BN}(\mathcal{T}) \leq T - 1$. Recall that $\mathcal{T}$ can be identified by a sequence of instances $x_1, \ldots, x_d$ and a sequence of functions $Y_1, \ldots, Y_d$ where $Y_t : \{0, 1\}^t \to \mathcal{Y}$ for every $t \in [d]$. For any path $\sigma \in \{0, 1\}^d$ down $\mathcal{T}$, the set $\{Y_t(\sigma_{\leq t})\}_{t=1}^d$ gives the labels along this path. Moreover, as all the branching on $\mathcal{T}$ occurs on levels in $S$, we have $\sum_{t=1}^d \mathbb{1}\{Y_t((\sigma_{<t}, 0)) \neq Y_t((\sigma_{<t}, 1))\} = \sum_{t \in S} \mathbb{1}\{Y_t((\sigma_{<t}, 0)) \neq Y_t((\sigma_{<t}, 1))\} = \lfloor \log T \rfloor$ for every path $\sigma \in \{0, 1\}^d$.

We now specify the stream to be observed by the learner $\mathcal{A}$. Draw $\sigma \sim \mathrm{Uniform}(\{0, 1\}^d)$ and consider the stream $\{(x_t, Y_t(\sigma_{\leq t}))\}_{t \in S}$. Repeat $(x_m, Y(\sigma_{\leq m}))$ for remaining $T - |S|$ timepoints, where $m$ is the largest index in set $S$. Since this stream is a sequence of instance-label pairs along the path $\sigma$ in the shattered tree $\mathcal{T}$, there exists a $c_\sigma \in \mathcal{C}$ consistent with the stream. However, using similar arguments as in the proof of the lowerbound $\mathrm{D}(\mathcal{C})/2$, we can establish

$$\mathbb{E}\left[\sum_{t=1}^T \mathbb{1}\{\mathcal{A}(x_t) \neq Y_t(\sigma_{\leq t})\}\right]$$

$$\geq \mathbb{E}\left[\sum_{t \in S} \mathbb{1}\{\mathcal{A}(x_t) \neq Y_t(\sigma_{\leq t})\}\right]$$

$$\geq \frac{1}{2} \mathbb{E}\left[\sum_{t \in S} \mathbb{1}\left\{Y_t((\sigma_{<t}, 0)) \neq Y_t((\sigma_{<t}, 1))\right\}\right]$$

$$= \frac{1}{2}\lfloor \log T \rfloor.$$

This completes our proof of lower bound.

## 4 Discussion

In this paper, we study the problem of multiclass transductive online learning with possibly arbitrary label space. In the realizable setting, we establish a trichotomy in the possible minimax rates of the expected number of mistakes. Furthermore, we show near-tight upper and lower bounds on the optimal expected regret in the agnostic setting. Along the way, we introduce two new combinatorial complexity parameters, called the Level-constrained Littlestone dimension and the Level-constrained Branching dimension.

Finally, we highlight some future directions of this work. First, can we extend our results to settings such as transductive online learning under bandit feedback, list transductive online learning, and transductive online real-valued regression? Moreover, as our shattering technique is general, can we use similar ideas to establish the possible minimax rates of the number of mistakes in the self-directed and the best-order settings initially studied in [Ben-David et al., 1995, 1997]?

## Acknowledgments and Disclosure of Funding

VR acknowledges support from the NSF Graduate Research Fellowship Program.

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

# A    Related Work

**Online Learning.** Online learning has been a subject of study for more than half a century. Moreover, the seminal work by Littlestone [1988] initiated this line of research within the computer science community. Since that pivotal contribution, online learning has been explored in various settings, from learning under the bandit feedback Daniely et al. [2011], Daniely and Helbertal [2013], Long [2017], Geneson [2021], Raman et al. [2023], Hanneke and Yang [2023] to quantum settings Aaronson et al. [2018], Mohan and Tewari [2023]. Furthermore, it is also linked to a broad set of problems, such as differential privacy, highlighted in studies by Alon et al. [2019], Bun et al. [2020], Alon et al. [2022]. Further, given its fundamental nature, it is not surprising that online learning has found numerous practical applications.

**Transductive and other Online Learning Frameworks.** The concept of the transductive learning model traces its origins to seminal works by Vapnik Vapnik and Chervonenkis [1974], Vapnik [1982, 1998], where it was explored within the PAC learning framework. Subsequently, Ben-David et al. [1997] initiated the study of this model under the umbrella of online learning, referring to it as "offline learning". They utilizes a notion of rank based dimension to prove their results. Notably, as we mentioned in the introduction, our Level-constrained Branching dimension is exactly equal to their rank based dimension. A significant advancement came recently with the work of Hanneke et al. [2023b]. Furthermore, other conceptually related models have also been rigorously studied, as seen in works by Goldman and Sloan [1994], Ben-David et al. [1995], Ben-David and Eiron [1998], Devulapalli and Hanneke [2024]. These studies notably include the self-directed online learning framework, which allows the learning algorithm to select the next instance for prediction from the remaining set of instances in each round, and additionally, the best order, which allows the learner (instead of an adversary) to select the order at the beginning of the game.

**Multiclass Classification.** A substantial volume of theoretical research has been conducted on various aspects of multiclass classification, as demonstrated by studies Natarajan and Tadepalli [1988], Natarajan [1989], Ben-David et al. [1992], Haussler and Long [1995], Rubinstein et al. [2006], Daniely et al. [2011, 2012], Daniely and Shalev-Shwartz [2014], Brukhim et al. [2021]. Despite this extensive body of work, a combinatorial characterization of multiclass classification with an infinite number of classes under Valiant's PAC learning framework in the realizable setting remained open until recently. In pursuit, the seminal paper by Brukhim et al. [2022] provided a combinatorial characterization in the mentioned setting. A key innovation in this breakthrough was the utilization of list learners. This dimension also serves to characterize the agnostic variant of this problem David et al. [2016]. For standard multiclass online learning with potentially unbounded label space, Daniely et al. [2011] presented a characterization for the realizable setting. Building on this, Hanneke et al. [2023a] extended Ben-David et al. [2009] technique to the agnostic setting with infinite label space. Notably, a similar trend can also be observed in the online learning under bandit feedback in the work of Daniely et al. [2011] followed by Raman et al. [2023].

# B    Proof of Proposition 1

Let $\mathcal{C} \subseteq \mathcal{Y}^{\mathcal{X}}$ be any concept class. To see that $\mathrm{D}(\mathcal{C}) \leq \mathrm{B}(\mathcal{C})$, note that if $\mathrm{D}(\mathcal{C}) = d$, there exists a Level-constrained Littlestone tree $\mathcal{T}$ of depth $d$ with branching factor $d$. Thus, it must be the case that $\mathrm{B}(\mathcal{C}) \geq d$.

To prove that $\mathrm{B}(\mathcal{C}) \leq \mathrm{L}(\mathcal{C})$, it suffices to show that for every shattered Level-constrained Branching tree $\mathcal{T}$ with branching factor $n \in \mathbb{N}$, there exists a shattered Littlestone tree of depth $n$. In particular, we will prove via induction the following claim: if $\mathcal{T}$ is a Level-constrained Branching tree with branching factor $n$ shattered by some $\mathcal{C}' \subseteq \mathcal{C}$, then there exists a Littlestone tree $\mathcal{T}'$ of depth $n$ shattered by $\mathcal{C}'$.

For the base case let $\mathcal{T}$ be a Level-constrained Branching tree with branching factor $1$ shattered by some $\mathcal{C}' \subseteq \mathcal{C}$. Without loss of generality (see Lemma 1), suppose that branching occurs on the root node of $\mathcal{T}$. Then, it is clear that just the root node of $\mathcal{T}$ along with its two outgoing edges is a Littlestone tree of depth $1$ shattered by $\mathcal{C}'$.

Now for the induction step, suppose the induction hypothesis is true for some $n \leq \mathrm{B}(\mathcal{C}) - 1$. Let $\mathcal{T}$ be a Level-constrained Branching tree with branching factor $n + 1$ shattered by $\mathcal{C}' \subseteq \mathcal{C}$. Again, without loss of generality, suppose branching occurs on the root node of $\mathcal{T}$. Let $\mathcal{T}_0$ and $\mathcal{T}_1$ be the left and right subtrees of $\mathcal{T}$ respectively shattered by $\mathcal{C}'_0 \subset \mathcal{C}'$ and $\mathcal{C}'_1 \subset \mathcal{C}'$ respectively. Then, note that $\mathcal{T}_0$ and $\mathcal{T}_1$ must both have a branching factor exactly $n$. Then, by the induction hypothesis, there exist Littlestone trees $\mathcal{T}'_0$ and $\mathcal{T}'_1$ of depth $n$ shattered by $\mathcal{C}'_0$ and $\mathcal{C}'_1$ respectively. Since branching occurs at the root node, the tree $\mathcal{T}'$ obtained by keeping the root node and its two outgoing edges of $\mathcal{T}$, but replacing $\mathcal{T}_0$ and $\mathcal{T}_1$ with $\mathcal{T}'_0$ and $\mathcal{T}'_1$ respectively, is a Littlestone tree of depth $n + 1$ shattered by $\mathcal{C}'_0 \cup \mathcal{C}'_1 \subseteq \mathcal{C}'$.

## C  Proof of Lowerbound $\frac{\mathrm{D}(\mathcal{C})}{2}$

*Proof.* Fix a transductive online learner $\mathcal{A}$. We will construct a randomized, hard realizable stream such that the expected number of mistakes made by $\mathcal{A}$ is at least $\frac{\mathrm{D}(\mathcal{C})}{2}$. Using the probabilistic method then gives the stated lowerbound.

Let $d := \mathrm{D}(\mathcal{C})$. Then, by Definition 2, there exists a sequence of instances $x_1, .., x_d \in \mathcal{X}^d$ and a sequence of functions $\{Y_t\}_{t=1}^d$ where $Y_t : \{0, 1\}^t \to \mathcal{Y}$ such that for every $\sigma \in \{0, 1\}^d$, the following holds:

(i) $Y_t(\sigma_{<t}, 0) \neq Y_t(\sigma_{<t}, 1)$ for all $t \in [d]$.

(ii) $\exists c_\sigma \in \mathcal{C}$ such that $c_\sigma(x_t) = Y_t(\sigma_{\leq t})$ for all $t \in [d]$.

Let $\sigma \sim \{0, 1\}^d$ a denote bitstring of length $d$ sampled uniformly at random and consider the stream $(x_1, Y_1(\sigma_{\leq 1})), ..., (x_d, Y_d(\sigma_{\leq d}))$. By the definition, there exists a concept $c_\sigma \in \mathcal{C}$ such that $c_\sigma(x_t) = Y_t(\sigma_{\leq t})$ for all $t \in [d]$. Moreover, observe that

$$
\begin{aligned}
\mathbb{E}\left[\sum_{t=1}^d \mathbb{1}\{\mathcal{A}(x_t) \neq Y_t(\sigma_{\leq t})\}\right] &= \sum_{t=1}^d \mathbb{E}\left[\mathbb{1}\{\mathcal{A}(x_t) \neq Y_t(\sigma_{\leq t})\}\right] \\
&= \sum_{t=1}^d \mathbb{E}\left[\mathbb{E}\left[\mathbb{1}\left\{\mathcal{A}(x_t) \neq Y_t\big((\sigma_{<t}, \sigma_t)\big)\right\} \,\Big|\, \sigma_{<t}\right]\right] \\
&\geq \frac{1}{2}\sum_{t=1}^d \mathbb{E}\left[\mathbb{E}\left[\mathbb{1}\left\{Y_t\big((\sigma_{<t}, 0)\big) \neq Y_t\big((\sigma_{<t}, 1)\big)\right\} \,\Big|\, \sigma_{<t}\right]\right] \\
&= \frac{1}{2}\mathbb{E}\left[\sum_{t=1}^d \mathbb{1}\left\{Y_t\big((\sigma_{<t}, 0)\big) \neq Y_t\big((\sigma_{<t}, 1)\big)\right\}\right] = \frac{d}{2}.
\end{aligned}
$$

where the first inequality follows from the fact that $\sigma_t \sim \mathrm{Uniform}(\{0, 1\})$. This completes the proof. $\qquad\square$

## D  Proof of Lemma 1

*Proof.* (of Lemma 1) If $\mathrm{depth}(\mathcal{T}) \leq 2^q - 1$, the claim holds trivially. So, we assume $\mathrm{depth}(\mathcal{T}) > 2^q - 1$. We now proceed by induction on $q$. For the base case $q = 1$, let $\mathcal{T}$ denote a level-constrained tree of $\mathrm{depth}(\mathcal{T}) > 1$ shattered by $\mathcal{C}$ that satisfies property (a) and (b) specified in Lemma 1. First, consider the case where the root node of $\mathcal{T}$ has branching. Since every path in $\mathcal{T}$ can have exactly 1 branching, there can be no further branching in $\mathcal{T}$. Next, consider the case when the root node of $\mathcal{T}$ is not the branching node and $\ell$ is the first level in $\mathcal{T}$ with branching. There must be $2^{\ell-1}$ nodes in this level, henceforth denoted by $\{v_i\}_{i=1}^{2^{\ell-1}}$. Moreover, denote $\mathcal{T}_{v_i}$ to be the corresponding subtree in $\mathcal{T}$ with $v_i$ as the root node. Since $\mathcal{T}$ satisfy property (b) and there are no branching nodes before level $\ell$, the subtrees $\{\mathcal{T}_{v_i}\}_{i=1}^{2^{\ell-1}}$ must be identical. Since all subtrees $\{\mathcal{T}_{v_i}\}_{i=1}^{2^{\ell-1}}$ have branching on the root node, there can be no further branching in these subtrees beyond the root node. Therefore, there cannot be any other levels $\ell' > \ell$ in $\mathcal{T}$ with branching node. This establishes that $\ell$ is the only level in $\mathcal{T}$ with at least one branching node. In either case, we have $\mathrm{BN}(\mathcal{T}) \leq 1 = 2^q - 1$.

Assume that Lemma 1 is true for some $q = n \in \mathbb{N}$. We now establish Lemma 1 for $q = n + 1$. To that end, let $\mathcal{T}$ be a level-constrained tree with branching factor $q = n + 1$ shattered by $\mathcal{C}$ that satisfies (a) and (b). Let $\ell \geq 1$ be the first level in $\mathcal{T}$ with at least one branching node, and $\{\mathcal{T}_i\}_{i=1}^{2^\ell - 1}$ be all the subtrees with its root node being a node on level $\ell$. As argued in the base case, all these subtrees must be identical. Thus, branching occurs on the same set of levels on all these subtrees, which implies that $\mathrm{BN}(\mathcal{T}) = \mathrm{BN}(\mathcal{T}_i)$ for all $i \in [2^{\ell - 1}]$. Let $\mathcal{T}_1^0$ and $\mathcal{T}_1^1$ denote the left and right subtree following the two outgoing edges from the root node of $\mathcal{T}_1$. Since, there is branching on the root-node of $\mathcal{T}_1$, we must have $\mathrm{BN}(\mathcal{T}_1) \leq 1 + \mathrm{BN}(\mathcal{T}_1^0) + \mathrm{BN}(\mathcal{T}_1^1)$. For each $i \in \{0, 1\}$, the subtree $\mathcal{T}_1^i$ is a level-constrained tree shattered by $\mathcal{C}$ that satisfies properties (a) and (b) for $q = n$. Using the inductive concept, we have $\mathrm{BN}(\mathcal{T}_1^i) \leq 2^n - 1$ for $i \in \{0, 1\}$. Therefore, combining everything

$$\mathrm{BN}(\mathcal{T}) = \mathrm{BN}(\mathcal{T}_1) \leq 1 + \mathrm{BN}(\mathcal{T}_1^0) + \mathrm{BN}(\mathcal{T}_1^1) \leq 1 + 2^n - 1 + 2^n - 1 = 2^{n+1} - 1.$$

This completes our induction step. □

## E Minimax Rates in the Agnostic Setting

We go beyond the realizable setting, and establish the minimax regret in the agnostic setting in terms of the Level-constrained Littlestone dimension.

**Theorem 4** (Regret bound). *For every concept class $\mathcal{C} \subseteq \mathcal{Y}^{\mathcal{X}}$ and $T \geq \mathrm{D}(\mathcal{C})$, we have*

$$\sqrt{\frac{T \, \mathrm{D}(\mathcal{C})}{8}} \leq \mathrm{R}^\star(T, \mathcal{C}) \leq \sqrt{T \, \mathrm{D}(\mathcal{C})} \, \log\Big(\frac{eT}{\mathrm{D}(\mathcal{C})}\Big).$$

We note that the upper- and lower-bounds in Theorem 4 are only off only by a factor logarithmic in $T$. We leave it as an open question to establish a matching upper- and lower-bounds.

*Proof.* (of upper bound in Theorem 4) To prove the upper bound, we will use the agnostic-to-realizable reduction from Hanneke et al. [2023a] to convert our realizable learner in Section 3.2 to an agnostic learner with the claimed upper bound on expected regret. By Theorem 4 in [Hanneke et al., 2023a], any conservative deterministic learner $\mathcal{A}$ with mistake bound $M$ can be converted into an agnostic learner with expected regret at most $\sqrt{T \, M \, \log\Big(\frac{eT}{M}\Big)}$. Although the proof by Hanneke et al. [2023a] only coverts the conservative Standard Optimal Algorithm to an agnostic learner, the arguments are general enough such that the conversion can be adapted for any conservative deterministic mistake-bound learner. By Theorem 3 and the proof in Section 3.2, there exists a conservative deterministic realizable learner with mistake bound at most $\mathrm{D}(\mathcal{C}) \log\left(\frac{eT}{\mathrm{D}(\mathcal{C})}\right)$. Using the realizable-to-agnostic conversion from Theorem 4 in [Hanneke et al., 2023a] with the conservative-version of the realizable learner in Section 3.2 gives an agnostic learner with expected regret at most

$$\sqrt{T \, \mathrm{D}(\mathcal{C}) \log\Big(\frac{eT}{\mathrm{D}(\mathcal{C})}\Big) \log\left(\frac{eT}{\mathrm{D}(\mathcal{C}) \log\left(\frac{eT}{\mathrm{D}(\mathcal{C})}\right)}\right)} \leq \sqrt{T \, \mathrm{D}(\mathcal{C})} \, \log\Big(\frac{eT}{\mathrm{D}(\mathcal{C})}\Big),$$

completing the proof. □

*Proof.* (of lower bound in Theorem 4) Our proof of lower bound is identical to the lower bound for the binary setting proved in [Hanneke et al., 2023b, Theorem 6.1], which is just a simple adaptation of standard lower bound technique from [Ben-David et al., 2009]. Thus, we only outline the sketch of the proof here.

Let $d = \mathrm{D}(\mathcal{C})$. Consider a sequence of instances $\{x_1^\star, \ldots, x_d^\star\} \subset \mathcal{X}$ and a sequence of functions $\{Y_i\}_{i=1}^d$ that is shattered by $\mathcal{C}$ according to Definition 2. Pick the largest odd number $k \in \mathbb{N}$ such that $kd \leq T$. First, the adversary reveals the instances $\{x_1, \ldots, x_T\}$ such that $x_t = x_1^\star$ for $t = 1, \ldots, k$, followed by $x_t = x_2^\star$ for $t = k + 1, \ldots, 2k$, and so forth. If $T > kd$, take $x_t = x_d^\star$ for all $t > kd$. As for labels, the adversary will first sample $(\sigma_1, \sigma_2, \ldots, \sigma_T) \in \mathrm{Uniform}(\{0, 1\}^T)$. Then, for $t = 1, \ldots, k$, the labels are selected as $y_t = Y_1(\sigma_t)$. For $t = k + 1, \ldots, 2k$, the labels are selected as

$y_t = Y_2((\bar{\sigma}_1, \sigma_t))$, where $\bar{\sigma}_1 = \mathbb{1}\left\{ \sum_{t=1}^{k} \mathbb{1}[\sigma_1 = 0] < \sum_{t=1}^{k} \mathbb{1}[\sigma_1 = 1] \right\}$ is the majority bit in the first block $t = 1, \ldots, k$. One can define $y_t$ for all $t > 2k$ analogously. For this stream, the label $y_t$ is essentially equivalent to the bit $\sigma_t \in \{0, 1\}$. Therefore, following the exact same arguments as in [Hanneke et al., 2023b, Theorem 6.1] establishes the lower bound of $\sqrt{Td/8}$. This completes the sketch of our proof. □

**Remark 1.** *Let $k \in \mathbb{N}$ be the number of classes. Let $\mathcal{C} \subseteq \{1, 2, \ldots, k\}^{\mathcal{X}}$ be a concept class. It is notable that for small number of classes $k$ (i.e. $k << 2^{(\log T)^2}$), the Natarajan bound that can be proved using the technique of Hanneke et al. [2023b] can be smaller than the upper bound in terms of the Level-constrained Littlestone dimension. However, for large $k$ (i.e. $k >> 2^{(\log T)^2}$), our upper bound in terms of $\mathrm{D}(C)$ can be better.*

# F   Comparisons to Existing Combinatorial Dimensions

In this section, we compare the Level-constrained Littlestone dimension 2 and the Level-constrained Branching dimension 3 to existing combinatorial dimensions in multiclass learning.

## F.1   Existing Combinatorial Dimensions

**Definition 4** ($i$-neighbour). *Let $f, g \in \mathcal{Y}^d$ for some $d \in \mathbb{N}$. For every $i \in [d]$, we say that $f$ and $g$ are $i$-neighbours if $f_i \neq g_i$ and $\forall_{j \in [d] \setminus \{i\}}\ f_j = g_j$.*

**Definition 5** (DS dimension Daniely and Shalev-Shwartz [2014]). *Let $\mathcal{C} \subseteq \mathcal{Y}^{\mathcal{X}}$ be a concept class. Let $S \in \mathcal{X}^d$ be a sequence for some $d \in \mathbb{N}$. We say that $S$ is DS-shattered by $\mathcal{C}$, if there exists $F \subseteq \mathcal{C}, |F| < \infty$ such that for all $f \in \{g \mid g \in \mathcal{Y}^d, \exists_{g \in F}\ \forall_{i \in [d]}\ g_i = f(S_i)\}$ and for all $i \in [d]$, $f$ has at least one $i$-neighbor. The DS dimension of $\mathcal{C}$, denoted $\mathrm{DS}(\mathcal{C})$, is the maximal size of a sequence $S \in \mathcal{X}^d$ for some $d \in \mathbb{N}$ that is DS-shattered by $\mathcal{C}$.*

**Definition 6** (Graph dimension). *Let $\mathcal{C} \subseteq \mathcal{Y}^{\mathcal{X}}$ be a concept class. Let $S \subseteq \mathcal{X}$. We say that $S$ is G-shattered by $\mathcal{C}$, if there exists an $f : S \to \mathcal{Y}$ such that for every $T \subseteq S$ there is a $g \in \mathcal{C}$ such that:*

$$\forall_{x \in T}\ g(x) = f(x) \text{ and } \forall_{x \in S-T}\ g(x) \neq f(x)$$

*The graph dimension of $\mathcal{C}$, denoted $\mathrm{G}(\mathcal{C})$, is the maximal cardinality of a set $S \subseteq \mathcal{X}$ that is G-shattered by $\mathcal{C}$.*

**Definition 7** (Natarajan Threshold dimension). *Let $\mathcal{C} \subseteq \mathcal{Y}^{\mathcal{X}}$ be a concept class. Let $S \in \mathcal{X}^d$ be a sequence for some $d \in \mathbb{N}$. We say that $S$ is NT-shattered by $\mathcal{C}$, if there exist $f, g : [d] \to \mathcal{Y}$ such that $\forall_{i \in [d]}\ f(i) \neq g(i)$, and there exists $(c_0, c_1, c_2, \ldots, c_d) \in \mathcal{C}^{d+1}$ such that for every $i \in [d+1], j \in [d]$:*

$$c_{i-1}(S_j) = \begin{cases} f(j), & j < i \\ g(j), & j \geq i \end{cases}$$

*The Natarajan Threshold dimension of $\mathcal{C}$, denoted $\mathrm{NT}(\mathcal{C})$, is the maximal size of a sequence $S \in \mathcal{X}^d$ for some $d \in \mathbb{N}$ that is NT-shattered by $\mathcal{C}$.*

## F.2   Comparison

It is easy to show that for every concept class $\mathcal{C} \subseteq \mathcal{Y}^{\mathcal{X}}$, its Natarajan dimension is always less than or equal to its DS dimension. Moreover, the work of Brukhim et al. [2022] demonstrated there exists a concept class $\mathcal{C} \subseteq \mathcal{Y}^{\mathcal{X}}$ for which the Natarajan dimension is 1 but $\mathrm{DS}(\mathcal{C}) = \infty$. Here, we show that for every concept class $\mathcal{C} \subseteq \mathcal{Y}^{\mathcal{X}}$, its DS dimension is always less than equal to its Level-constrained Littlestone dimension. Furthermore, we demonstrate there exists a concept class $\mathcal{C}' \subseteq \mathcal{Y}^{\mathcal{X}}$ such that $\mathrm{DS}(\mathcal{C}') = 1$ but $\mathrm{D}(\mathcal{C}') = \infty$. These two results are shown in Proposition 2.

**Proposition 2.** *For every concept class $\mathcal{C} \subseteq \mathcal{Y}^{\mathcal{X}}$, we have: $\mathrm{DS}(\mathcal{C}) \leq \mathrm{D}(\mathcal{C})$. Moreover, there exists a concept class $\mathcal{C}' \subseteq \mathcal{Y}^{\mathcal{X}}$ such that $\mathrm{DS}(\mathcal{C}') = 1$ but $\mathrm{D}(\mathcal{C}') = \infty$.*

*Proof.* First, we prove that for every concept class $\mathcal{C} \subseteq \mathcal{Y}^{\mathcal{X}}$, we have: $\mathrm{DS}(\mathcal{C}) \leq \mathrm{D}(\mathcal{C})$. Let $\mathcal{C} \subseteq \mathcal{Y}^{\mathcal{X}}$ be a concept class such that $\mathrm{DS}(\mathcal{C})$ is finite. Subsequently, we show that we can construct a Level-constrained Littlestone tree $\mathcal{T}$ of depth $\mathrm{DS}(\mathcal{C})$, which is shattered by $\mathcal{C}$. Thus, $\mathrm{DS}(\mathcal{C}) \leq \mathrm{D}(\mathcal{C})$.

Let $S \in \mathcal{X}^{\mathrm{DS}(\mathcal{C})}$ be a sequence of instances, which is *DS-shattered* by $\mathcal{C}$. We show that we can construct a Level-constrained Littlestone tree $\mathcal{T}$ of depth $\mathrm{DS}(\mathcal{C})$, having members of $S$ as its nodes in order with the first member being its root and so on, which is shattered by $\mathcal{C}$. To show the construction, we use induction. If $\mathrm{DS}(\mathcal{C}) = 1$, it is clear that we can construct a Level-constrained Littlestone tree $\mathcal{T}$ of depth 1, which is shattered by $\mathcal{C}$. This is because there must be two concepts in $\mathcal{C}$, which disagree on one member of $S$. We assume that if $\mathrm{DS}(\mathcal{C}) = d$, we can construct a Level-constrained Littlestone tree $\mathcal{T}$ of depth $d$, having members of $S$ as its nodes in order with the first member being its root and so on, which is shattered by $\mathcal{C}$, where $S \in \mathcal{X}^d$ is a sequence of size $d$ witnessing $\mathrm{DS}(\mathcal{C}) = d$. Now, we prove that if $\mathrm{DS}(\mathcal{C}) = d + 1$, we can construct a Level-constrained Littlestone tree $\mathcal{T}$ of depth $d + 1$, having members of $S$ as its nodes in order with the first member being its root and so on, which is shattered by $\mathcal{C}$, where $S \in \mathcal{X}^{d+1}$ is a sequence of size $d + 1$ witnessing $\mathrm{DS}(\mathcal{C}) = d + 1$. Let $F \subset \mathcal{C}$ be a set witnessing $\mathrm{DS}(\mathcal{C}) = d + 1$. Take any two distinct concepts $c_1, c_2 \in F$. Define $F'$ as follows: $F' := \{f \mid f \in F, f(S_1) = c_1(S_1)\}$. In addition, define $F''$ as follows: $F'' := \{f \mid f \in F, f(S_1) = c_2(S_1)\}$. Observe that $(S_2, S_3, \ldots, S_{d+1})$ and $F'$ can witness $\mathrm{DS}(\mathcal{C}) \geq d$. Similarly, observe that $(S_2, S_3, \ldots, S_{d+1})$ and $F''$ can witness $\mathrm{DS}(\mathcal{C}) \geq d$. Now, we set the root of $\mathcal{T}$ as $S_1$ and branches with $c_1(S_1)$ and $c_2(S_1)$ labels. Based on the inductive assumption combined with the facts that we mentioned, we can complete the construction of Level-constrained Littlestone tree $\mathcal{T}$ of depth $\mathrm{DS}(\mathcal{C})$, having members of $S$ as its nodes in order with the first member being its root and so on, which is shattered by $\mathcal{C}$.

Finally, we note that if $\mathrm{DS}(\mathcal{C}) = \infty$, as we can do the construction for every depth $d \in \mathbb{N}$, we should have $\mathrm{D}(\mathcal{C}) = \infty$.

Second, we prove that there exists a concept class $\mathcal{C}' \subseteq \mathcal{Y}^{\mathcal{X}}$ such that $\mathrm{DS}(\mathcal{C}') = 1$ but $\mathrm{D}(\mathcal{C}') = \infty$. To show this, we use our next proposition, namely 3, combined with the well-known fact that for every $\mathcal{C} \subseteq \mathcal{Y}^{\mathcal{X}}$, we have: $\mathrm{DS}(\mathcal{C}) \leq \mathrm{G}(\mathcal{C})$. $\qquad \square$

Next, we show there that exists a concept class $\mathcal{C} \subseteq \mathcal{Y}^{\mathcal{X}}$ such that $\mathrm{G}(\mathcal{C}) = 1$ and $\mathrm{D}(\mathcal{C}) = \infty$. On the other hand, we also prove the existence of a concept class $\mathcal{C} \subseteq \mathcal{Y}^{\mathcal{X}}$ such that $\mathrm{G}(\mathcal{C}) = \infty$ and $\mathrm{D}(\mathcal{C}) = 1$. These two results, shown in Proposition 3, imply that the Level-constrained Littlestone dimension and the Graph dimension are not comparable. Moreover, our first claim has an interesting consequence. In particular, it illustrates that having a finite Level-constrained Littlestone dimension is not necessary for having a bounded size sample compression scheme. This follows from the fact that having finite Graph dimension is sufficient for having a bounded size sample compression scheme [David et al., 2016]. We also remark that for every concept class $\mathcal{C} \subseteq \mathcal{Y}^{\mathcal{X}}$, its DS dimension is always less than or equal to its Graph dimension.

**Proposition 3.** *There exists a concept class $\mathcal{C} \subseteq \mathcal{Y}^{\mathcal{X}}$ such that $\mathrm{G}(\mathcal{C}) = 1$ and $\mathrm{D}(\mathcal{C}) = \infty$. Moreover, there exists a concept class $\mathcal{C}' \subseteq \mathcal{Y}^{\mathcal{X}}$ such that $\mathrm{G}(\mathcal{C}') = \infty$ and $\mathrm{D}(\mathcal{C}') = 1$.*

*Proof.* First, we prove the second claim. To show that, we rely on Example 1 in Hanneke et al. [2023a]. In particular, they showed there exists a concept class $\mathcal{C}' \subseteq \mathcal{Y}^{\mathcal{X}}$ such that $\mathrm{G}(\mathcal{C}') = \infty$ and $\mathrm{L}(\mathcal{C}') = 1$. As we know $\mathrm{D}(\mathcal{C}') \leq \mathrm{L}(\mathcal{C})$, we conclude there exists a concept class $\mathcal{C}' \subseteq \mathcal{Y}^{\mathcal{X}}$ such that $\mathrm{G}(\mathcal{C}') = \infty$ and $\mathrm{D}(\mathcal{C}') = 1$.

Second, we prove that there exists a concept class $\mathcal{C} \subseteq \mathcal{Y}^{\mathcal{X}}$ such that $\mathrm{G}(\mathcal{C}) = 1$ and $\mathrm{D}(\mathcal{C}) = \infty$. Let $\mathcal{T}$ be an infinite depth rooted perfect binary tree so that all of its levels and edges are labeled by distinct elements. The definition of such a tree is similar to Definition 1.7 in the work of Bousquet et al. [2021]. Let $\mathcal{X}$ be the elements on the levels of $\mathcal{T}$ and $\mathcal{Y}$ be the elements on the edges of $\mathcal{T}$. Also, define the concept class $\mathcal{C} \subseteq \mathcal{Y}^{\mathcal{X}}$ as follows: $\mathcal{C}$ only contains all concepts consistent with a branch of $\mathcal{T}$. Thus, clearly, we have: $\mathrm{D}(\mathcal{C}) = \infty$. Now, we show that $\mathrm{G}(\mathcal{C}) = 1$. We prove this by contradiction. Assume $\mathrm{G}(\mathcal{C}) \geq 2$. Thus, there exist $S = (x_1, x_2) \subset \mathcal{X}$ of size 2 and $f : S \to \mathcal{Y}$ witnessing the fact that $\mathrm{G}(\mathcal{C}) = 2$. Without loss of generality, we assume that $x_1$ is above $x_2$ in $\mathcal{T}$. Using the fact that the edges of $\mathcal{T}$ are labeled with distinct elements of $\mathcal{Y}$, there cannot exist both $c_1 \in \mathcal{C}$ and $c_2 \in \mathcal{C}$ such that $c_1(x_1) = f(x_1)$, $c_1(x_2) = f(x_2)$, $c_2(x_1) \neq f(x_1)$, and $c_2(x_2) = f(x_2)$. This is a contradiction, thus $\mathrm{G}(\mathcal{C}) = 1$. $\qquad \square$

It is well-known that for every concept class $\mathcal{C} \subseteq \mathcal{Y}^{\mathcal{X}}$, its Littlestone dimension is always less than equal to its sequential graph dimension. Moreover, the work of Hanneke et al. [2023a] demonstrated there exists a concept class $\mathcal{C} \subseteq \mathcal{Y}^{\mathcal{X}}$ such that $\mathrm{L}(\mathcal{C}) = 1$ and $\mathrm{SG}(\mathcal{C}) = \infty$. Here, we show that for

every concept class $\mathcal{C} \subseteq \mathcal{Y}^{\mathcal{X}}$, its Level-constrained Branching dimension is always less than equal to its Littlestone dimension. Furthermore, we demonstrate that there exists a concept class $\mathcal{C}' \subseteq \mathcal{Y}^{\mathcal{X}}$ such that $B(\mathcal{C}') \leq 2$ and $L(\mathcal{C}') = \infty$. These two results are shown in Proposition 4.

**Proposition 4.** *For every concept class $\mathcal{C} \subseteq \mathcal{Y}^{\mathcal{X}}$, we have:* $B(\mathcal{C}) \leq L(\mathcal{C})$. *Moreover, there exists a concept class $\mathcal{C}' \subseteq \mathcal{Y}^{\mathcal{X}}$ such that $B(\mathcal{C}') \leq 2$ and $L(\mathcal{C}') = \infty$.*

*Proof.* The proof of the following claim: for every concept class $\mathcal{C} \subseteq \mathcal{Y}^{\mathcal{X}}$, we have that $B(\mathcal{C}) \leq L(\mathcal{C})$ is given by Proposition 1. Therefore, we focus on showing that there exists a concept class $\mathcal{C}' \subseteq \mathcal{Y}^{\mathcal{X}}$ such that $B(\mathcal{C}') \leq 2$ and $L(\mathcal{C}') = \infty$. Let $\mathcal{T}$ be an infinite depth rooted perfect binary tree so that all of its nodes are labeled by distinct elements, all of its left edges are labeled by 0, and all of its right edges are labeled by 1. The definition of such a tree is similar to Definition 1.7 in the work of Bousquet et al. [2021]. Let $\mathcal{X}$ be the elements on the nodes of $\mathcal{T}$. Also, define the concept class $\mathcal{C}'$ as follows: $\mathcal{C}'$ contains only the concepts consistent with a branch of $\mathcal{T}$. Further, each of these concepts predicts a unique label for all instances outside its associated branch. In addition, define $\mathcal{Y}$ as the union of $\{0, 1\}$ and all unique labels used in the definition of $\mathcal{C}'$. Thus, we have: $L(\mathcal{C}') = \infty$. Now, we show that $B(\mathcal{C}') \leq 2$. To prove this, we demonstrate that for every $T \in \mathbb{N}$, we have: $\inf_{\text{Deterministic } \mathcal{A}} M_{\mathcal{A}}(T, \mathcal{C}') \leq 2$. As a result, we can then conclude that $B(\mathcal{C}') \leq 2$.

To see why $\inf_{\text{Deterministic } \mathcal{A}} M_{\mathcal{A}}(T, \mathcal{C}') \leq 2$ for every $T \in \mathbb{N}$ implies that $B(\mathcal{C}') \leq 2$, suppose for the sake of contradiction that $B(\mathcal{C}') \geq 3$. So, there exists a Level-constrained Branching tree of depth $d \in \mathbb{N}$ such that its branching factor is at least 3. Let $T' = d$. It is not hard to see that there exists a sequence of instances of size $T'$ such that for every deterministic learner, there exists a realizable labeling of instances that forces the learner to make at least 3 mistakes over $T'$ rounds. This leads to a contradiction. Thus, we conclude that $B(\mathcal{C}') \leq 2$.

We now construct a deterministic learner $\mathcal{A}$ such that $M_{\mathcal{A}}(T, \mathcal{C}') \leq 2$ for every $T \in \mathbb{N}$. Let $T \in \mathbb{N}$. Let $S\mathcal{X}^T$ be the sequence chosen by the adversary at the beginning of the game. Also, let $c^\star \in \mathcal{C}'$ be the target concept chosen by the adversary. Further, let $u$ be the root-to-leaf path in $\mathcal{T}$ associated with the concept $c^\star$. In addition, for every $i \in [T]$, let $v_i$ be a root-to-leaf path in $\mathcal{T}$ containing first $i$ members of $S$, if it exists. Finally, let $i^\star$ be the smallest positive integer such that $v_{i^\star}$ does not exist. If $i^\star$ itself does not exist, let $i^\star = T + 1$.

Our algorithm $\mathcal{A}$ predicts according to the $\{0, 1\}$ labels associated with the path $v_{i^\star - 1}$ for the first $i^\star - 1$ points in $S$. Furthermore, if the adversary ever reveals a unique label, we use its corresponding $c \in \mathcal{C}'$ to make predictions in all future rounds. For the $i^\star$'th member of $S$, if it exists, we predict arbitrarily. To see that this algorithm makes at most 2 mistakes, we consider two cases. (1) If $i^\star = T + 1$, then our algorithm makes at most one mistake. In fact, our algorithm makes a mistake: (a) if the adversary switches the label from a bit in $\{0, 1\}$ to a unique label corresponding to the target concept $c^\star$. (b) perhaps on the last instance. (2) Otherwise, the algorithm makes at most two mistakes; the first mistake can be on round $i^\star - 1$, and the second mistake can be on round $i^\star$, after which the true $c^\star$ is known to the learner from its unique label. Indeed, if the adversary switches the label from a bit in $\{0, 1\}$ to a unique label corresponding to the target concept $c^\star$ before round $i^\star - 1$, we only make one mistake. This completes the proof. □

Finally, the works of Shelah [1990], Hodges [1997] showed that the finiteness of the Littlestone and Threshold dimensions coincide in the binary setting. Here, we show that this is not the case between the Level-constrained Branching dimension and the Natarajan Threshold dimension. More specifically, we show that for every concept class $\mathcal{C} \subseteq \mathcal{Y}^{\mathcal{X}}$, its Level-constrained Branching dimension is always greater than or equal to the log of its Natarajan Threshold dimension. However, we give a concept class $\mathcal{C}' \subseteq \mathcal{Y}^{\mathcal{X}}$ such that $NT(\mathcal{C}') = 1$ and $B(\mathcal{C}') = \infty$. These two results are shown in Proposition 5. Notably, the lower bound of Hanneke et al. [2023b], based on the threshold dimension, can be easily generalized to our setting for the Natarajan Threshold dimension.

**Proposition 5.** *For every concept class $\mathcal{C} \subseteq \mathcal{Y}^{\mathcal{X}}$, we have:* $\log(NT(\mathcal{C})) \leq B(\mathcal{C})$. *Moreover, there exists a concept class $\mathcal{C}' \subseteq \mathcal{Y}^{\mathcal{X}}$ such that $NT(\mathcal{C}') = 1$ and $B(\mathcal{C}') = \infty$.*

*Proof.* First, we prove that for every concept class $\mathcal{C} \subseteq \mathcal{Y}^{\mathcal{X}}$, we have: $\log(NT(\mathcal{C})) \leq B(\mathcal{C})$. Let $\mathcal{C} \subseteq \mathcal{Y}^{\mathcal{X}}$ be a concept class such that $NT(\mathcal{C}) = d$ for some $d \in \mathbb{N}$. Let $T = d$. On the one hand, by presenting the sequences of instances that are NT-shattered by $\mathcal{C}$ to the learner, we can use a similar technique as [Hanneke et al., 2023b, Claim 3.4], to prove a lower bound of $\log(NT(\mathcal{C}))$ on $M^\star(T, \mathcal{C})$.

On the other hand, based on Section 3, we can prove an upper bound of $B(\mathcal{C})$ on $M^\star(T, \mathcal{C})$. Thus, we have $\log(\mathrm{NT}(\mathcal{C})) \leq B(\mathcal{C})$.

Second, we prove that there exists a concept class $\mathcal{C}' \subseteq \mathcal{Y}^\mathcal{X}$ such that $\mathrm{NT}(\mathcal{C}') = 1$ and $B(\mathcal{C}') = \infty$. Let $\mathcal{T}$ be a rooted binary tree so that it has the following three properties: (1) all of its levels and edges are labeled by distinct elements. (2) each level only contains one node with two children (3) its branching factor is infinite. It is not hard to see that such a tree exists. The definition of such a tree is similar to Definition 1.7 in the work of Bousquet et al. [2021]. Let $\mathcal{X}$ be the elements on the levels of $\mathcal{T}$ and $\mathcal{Y}$ be the elements on the edges of $\mathcal{T}$. Also, define the concept class $\mathcal{C}' \subseteq \mathcal{Y}^\mathcal{X}$ as follows: $\mathcal{C}'$ only contains all concepts consistent with a branch of $\mathcal{T}$. Thus, clearly, we have: $B(\mathcal{C}') = \infty$. Now, we show that $\mathrm{NT}(\mathcal{C}') = 1$. We prove this by contradiction. Assume $\mathrm{NT}(\mathcal{C}') \geq 2$. Then, there exist $S = (x_1, x_2) \in \mathcal{X}^2$ and $(c_0, c_1, c_2) \in \mathcal{C}'^3$ witnessing $\mathrm{NT}(\mathcal{C}') = 2$. Without loss of generality, we assume that $x_1$ is above $x_2$ in $\mathcal{T}$. Based on our constriction of $\mathcal{T}$, it is simple to see that $c_0(x_2) \neq c_1(x_2)$ and $c_0(x_2) \neq c_2(x_2)$ and $c_1(x_2) \neq c_2(x_2)$. Thus, $\mathrm{NT}(\mathcal{C}')$ can not be even 2, which completes our contradiction-based proof. $\qquad\square$

