# OpenReview forum: "Multiclass Transductive Online Learning"
_NeurIPS.cc/2024/Conference — NeurIPS 2024 spotlight_

### Official Review · Reviewer_Nc66 · 2024-06-17

**Soundness:** 3
**Presentation:** 3
**Contribution:** 3
**Rating:** 8
**Confidence:** 4

**Summary:**

This paper studies transductive online learning in the multiclass setting, where the label space can be unbounded. In the transductive setting, the adversary commits to a sequence of examples and can only adaptively choose labels for a sequence of instances, unlike the pure online setting where the adversary can adaptively choose both the sequence of examples and the labels, or the pure offline where the adversary commits to both a sequence of examples and labels.

The main result of this paper extends the results of Hanneke et. al. [2023] by characterizing the optimal mistake bound in the unbounded label case. The proof techniques involve defining and leveraging two new combinatorial parameters termed Level-constrained Branching and Littlestone dimensions. The authors prove that the level-constrained Littlestone dimension characterizes this specific variant of online learning.


Hanneke et. al. [2023]: A Trichotomy for Transductive Online Learning.

**Strengths:**

1. This paper characterizes the optimal mistake bound of transductive online learners in the unbounded label space setting in both realizable and agnostic cases.

2. The paper introduces two new combinatorial parameters: the Level-constrained Littlestone dimension $D(C)$ and the Level-constrained Branching dimension $B(C)$, and show that $D(C)$ characterizes transductive online learning in the unbounded label setting. These new parameters are also compared to existing combinatorial parameters like the Natarajan Dimension, DS dimension, and the Graph dimension.

3. The technical parts of the paper involve modifying the Halving technique to work with  new notion of shattering using the above newly introduced combinatorial parameters,  and a modified version of Littlestone's standard optimal algorithm. Combined with the definition of $D(C)$ and $B(C)$, I feel that these new definitions and techniques could be of independent interest to the learning theory community.

4. This work almost rounds out the literature on regret and mistake bounds in the original transductive online learning setting. Removing the logarithmic factor in the upper bound of Theorem 3 seems to be the final milestone in this setting.

**Weaknesses:**

1. The definitions of expected regret and expected cumulative mistakes in section 2.2 seem to be incomplete. What is the probability distribution that the expectations are taken with respect to? The only clear choice to me is some probability distribution over the label space $\mathcal{Y}$. Since the label space is unbounded (a critical point of this work), the notion of a probability distribution over this unbounded label space should be rigorously defined somewhere in the paper (maybe in the Appendix if there are space constraints). This seems to be an inexplicable oversight in an otherwise highly technical and detailed paper.

2. Even though the regret in the agnostic setting is stated in terms of the level-constrained Littlestone dimension, the proof technique heavily borrows from previous works. The authors state this clearly in the text as well. While the result serves a pedagogical purpose and also lends to completeness of the results, Theorem 4 cannot be counted as a significant contribution.

The following weaknesses do not directly impact my scores but should improve the paper's readability.

3. In reference [Brukhim 2022], the DS dimension characterizes multiclass learnability for unbounded label space in the offline setting. It is not immediately clear from the main text why the DS dimension cannot similarly characterize the transductive online setting and why $D(C)$ and $B(C)$ are required in the first place. The answer is possibly buried in Proposition 2, given in Appendix F.2, and should clearly be stated in the main text (even informal statements suffice), given the fact that the new combinatorial parameters are a core contribution of this paper.

4. The proofs in section 3.1 and 3.2 use the standard Halving technique, but make use of the newly defined combinatorial parameters. Both upper bounds share similar overarching ideas with each other, and also with results in previous works for the constrained label space. These two sections can be rewritten in a manner that highlights the power of the Halving technique. While the proofs themselves appear to be correct, if we take the above weakness into consideration, its also not immediately apparent from the main text why the Halving technique cannot be applied using more standard combinatorial parameters such as the DS dimension. Also, I feel that some parts of the proof can be deferred to the appendix (up to the discretion of the authors).

**Questions:**

Minor issues
------------------
1. It is unclear from the definitions in section 2.2 how optimal regret and optimal mistake bound would be defined differently in the realizable setting. Could the authors expand on this?

2. Equations on lines 210, 285, and 354 should be formatted properly.

Future Work Discussions
-----------------------------------
Please feel free not to answer these questions if the authors are cramped for time. Not answering these questions will not affect my score.

1. Do the lower bounds of Theorem 3 hold directly for the list transductive online setting?
2. In the online real-valued regression setting, can one reduce a discretized version of the problem to the unbounded label case setting?

**Limitations:**

Adequately discussed in the paper.

---

> ### Author Rebuttal · Authors · 2024-08-05
>
> We thank the reviewer for pointing out that our techniques could be of independent interest to the learning theory community and that our result almost rounds out the literature on regret and mistake bounds in the original transductive online learning setting. All minor issues and suggestions will be incorporated in the final version. We address each weakness and question below.
>
> - The expectation is taken with respect to only the randomness of the learner. The learner makes predictions by sampling from distributions over $Y$. Thus, the reviewer is correct in the sense that the expectation is taken with respect to distributions over the label space $Y$. Regarding issues of measurability, following the work of [1], we only require that the singleton sets $\lbrace{y \rbrace}$ need to be measurable. We will make the assumption needed on $Y$ more clear in the camera-ready version.
> - We agree with the reviewer that our agnostic upper bound uses a pretty standard technique. Our main contribution is in the realizable setting.
> - Hanneke et al. [2023] in Claim 5.3 show that the DS dimension cannot characterize multiclass transductive online learnability. Namely, they give a class $C$ such that $DS(C) = 1$, but $C$ is not transductive online learnable. This necessitates the need for new dimensions since the Littlestone dimension is clearly not necessary. We will make this explicit in the main text of the camera ready version.
> - We provided a brief explanation of the relevance of the Halving algorithm in transductive online learning in lines 110-120. In particular, we noted that a naive adaptation of the Halving algorithm for multiclass learning would not work and that we defined a new notion of shattering that would allow us to apply an analog of the Halving algorithm. Nevertheless, we will make sure to point out that both algorithms in Section 3.1 and 3.2 share similar overarching ideas in the sense that they both use the Halving technique with a particular combinatorial dimension. We note that the Halving technique using the DS dimension cannot work because the finiteness of the DS dimension is not sufficient for online learnability.
> - In the realizable setting, we evaluate the learner through its mistake bound. On the other hand, in the agnostic setting, we evaluate the learner through its regret bound. In the realizable setting, the regret bound and mistake bound are the same quantities. However, this is not the case in the agnostic setting, where one usually only cares about the regret bound.
> - Regarding the list setting, our lower bound applies to the list transductive online setting. However, it is possible to establish tighter lower bounds by adapting our definitions for $(L+1)$-array trees when the list size is $L$. Regarding the regression setting, one could approach this problem using binary results. We are uncertain whether our results will be directly applicable.
>
> [1] S. Hanneke, S. Moran, V. Raman, U. Subedi, A. Tewari. Multiclass Online Learning and Uniform Convergence. 36th Conference on Learning Theory, 2023.

---

> > ### Comment · Reviewer_Nc66 · 2024-08-08
> >
> > Thank you for the response and for an excellent paper! I am thoroughly satisfied with the responses and I will be raising my score further in order to stress the contribution of this work.

---

### Official Review · Reviewer_DRE4 · 2024-07-10

**Soundness:** 3
**Presentation:** 3
**Contribution:** 3
**Rating:** 7
**Confidence:** 4

**Summary:**

The paper studies the problem of multiclass transductive online learning where the number of classes can be unbounded.

In the transductive setting, the learner receives in advance a sequence of instances $(x_1,…,x_T)$ by an adversary. Then, sequentially at each time step t, it needs to decide a label $\hat{y}_t$, and then the adversary reveals the true label label $y_t$. Given a concept class $C$, in the realizable setting, the adversary must choose the labels according to a concept $c \in C$.  The goal is to design an online algorithm that minimizes the regret, i.e. the total number of mistakes done by the learner compared to the best concept chosen from C in hindsight.

The paper fully characterizes the regret for the realizable and the agnostic setting for this problem in the case of an infinite number of classes, extending previous results of Hanneke et al. [2024] limited to a finite number of classes.

**Strengths:**

The paper characterizes the multiclass transductive online learning problem in the case of an infinite number of classes. This is a nice contribution to a fundamental problem. The paper is well written.

**Weaknesses:**

It is not clear whether the tools presented in this paper can be applied to other settings. Specifically, the definitions of level-constrained branching dimension, and level-constrained littlestone dimension seem specific to solve this (still important) problem. (See also question 2 below).

In Lines 102-103, it is claimed that finiteness of D(C) and B(C) coincides for |Y| = 2. However, if that’s the case, it seems that Theorem 1 cannot show the Trichotomy of Hanneke et al 2024 for the binary case.

**Questions:**

1. Does the label set Y need to be countable?

2. The DS dimension characterizes the learnability of multiclass learning in the pac setting (Brukhim et al 2022) . Would it be possible to express Theorem 1 using DS and D (i.e., can DS replace the branching dimension, to have a nice parallel between VC and DS?)

Typos
207 upper=
211 upper round
359 lowerbounds

**Limitations:**

There is no specific section for limitations. This theory paper is self-contained so I do not believe it is needed.

---

> ### Author Rebuttal · Authors · 2024-08-05
>
> We thank the reviewer for finding our work to be well written and a nice contribution to a fundamental problem. All minor typos and suggestions will be incorporated in the final version. We address each weakness and question below.
>
> - The algorithm achieving the $\log{T}$ upper bound in the realizable setting is the most significant contribution of our work, as mentioned in Section 1.2. We believe that the adaptation of this algorithm is applicable to various other settings, including list transductive online learning and transductive online learning with bandit feedback.
> - In lines 102-103, we first stated that $D(C) = VC(C)$ for binary classification. Then, we stated that $B(C) < \infty$ if and only if $L(C) < \infty$ for binary classification. We did not make the claim that $B(C) < \infty$ if and only if $D(C) < \infty$.
> - The label space $Y$ does not need to be countable. Following the work of [1], we only require that the singleton sets $\lbrace{y \rbrace}$ need to be measurable. We will make the assumption needed on $Y$ more clear in the camera-ready version.
> - No, it is not possible to express Theorem 1 using DS and D because $DS(C) \leq D(C)$ for every $C$. In addition, Claim 5.4 in Hanneke et al. [2023] shows that the DS dimension does not characterize multiclass transductive online learning. Namely, there is a class where $DS(C) = 1$, but $C$ is not transductive online learnable. However, if $|Y| < \infty$, then the DS dimension does characterize learnability. We also note that we have comparisons to other existing dimensions in Appendices B and F.
>
> [1] S. Hanneke, S. Moran, V. Raman, U. Subedi, A. Tewari. Multiclass Online Learning and Uniform Convergence. 36th Conference on Learning Theory, 2023.

---

> > ### Comment · Reviewer_DRE4 · 2024-08-11
> >
> > Thank you for your clarification and your response, and for pointing out to the appendix sections relevant to my questions.
> > (I am sorry for misreading Lines 102-103).
> >
> > After reading the other review and the rebuttal,  I still believe this is a good theory paper (and I increased my score accordingly).

---

### Official Review · Reviewer_Cx7F · 2024-07-12

**Soundness:** 4
**Presentation:** 3
**Contribution:** 3
**Rating:** 7
**Confidence:** 4

**Summary:**

This work continues the study of transductive online classification (a learning setting from the 90s recently reviewed by Hanneke et al. [NeurIPS 2023]. The main result is a trichotomy of possible rates for the general multi-class case (even for the infinite label case) in the realizable setting; answering an open question by Hanneke et al. The three cases are characterized by novel combinatorial dimensions (variant of Littlestone dimension called level-wise Littlestone and ).

Additionally they achieve optimal $\tilde{\Theta}(\sqrt{TD(\mathcal{C})})$ (up to log factors) rates in the agnostic setting again determined by the level-wise Littlestone dim. $D(\mathcal{C})$.

**Strengths:**

Timely and interesting paper continuing the recent interest in transductive online classification and related models. Almost tightly characterizes the possible rates for agnostic and realizable learning.

--- rebuttal ---
changed from 6 to 7

**Weaknesses:**

Not in-depth discussion of previous related work.
The gap between $D(C)$ and $B(C)$ in Theorem 3 (in the $B(C)<\infty$ case) is somewhat unsatisfactory.

See also questions and limitations below.

Minor:
* You might want to fix "Hanneke et al. [2024]" to [2023], otherwise you cite a NeurIPS paper in the future.
* typo in line 37: should be $c:X\to Y$ not $c:X\mapsto Y$.
* typo in line 207: "upper="
* Perhaps hint the additional results (Prop 4, comparison to DS, graph-dim) in the main paper, at least with some short sentences.

**Questions:**

In the agnostic setting for $|Y|=k$ one would expect something like $O(\sqrt{T\mathrm{Ndim}(C)k})$ up to log factors, similar to the bound in the realizable setting in Hanneke et al. 2023  (Theorem B.3). Please relate this to your agnostic bound.

Can the gap between $D(C)$ and $B(C)$ be arbitrary? More generally how far can $D(C)$,$B(C)$, and $L(C)$ be from each other. I agree that stating the rate as $O(1)$ makes sense, but it would be nice to have some quantity explicitly giving the rate in this case some Xdim s.t. $O(\mathrm{Xdim}(H))$ (as Ldim does for standard online classification). Note that the tree ranks by Ben-David et al [1997] (see limitations, as well), achieves this specifically also for the worst-sequence/transductive setting (at least for the binary case).

**Limitations:**

Please discuss previous work more thoroughly. The tree ranks from the "Online learning versus offline learning" (self-directed, worst-case sequence, etc.) [Ben-David et al. 1997] and similar papers are very much related to the proposed dimensions here (which is only very briefly acknowledged in a short sentence here). E.g., the "level-constrained" variant of the trees in these papers also become the VC-dim for $|Y|=2$. Also similarities to Devulapalli and Hanneke could be discussed more (e.g., the lower bounds there probably apply here too).

---

> ### Author Rebuttal · Authors · 2024-08-05
>
> We thank the reviewer for finding our work timely and interesting.  All minor typos and suggestions will be incorporated in the final version. We address each weakness and question below.
>
> - For small $k$ (i.e. $k << 2^{(\log{T})^2}$), the Natarajan bound can be smaller than the upper bound in terms of the Level-constrained Littlestone dimension. However, for large $k$ (i.e. $k > 2^{(\log{T})^2}$), our upper bound in terms of $D(C)$ can be better. We will make sure to point this out in the camera ready version.
> - There can indeed be an arbitrary gap between $B(C)$ and $D(C)$. For example, for the class of thresholds $C = \lbrace{x \mapsto 1\lbrace{x \geq a \rbrace}: a \in \mathbb{R} \rbrace}$, we have that $D(C) = VC(C) = 1$, however $B(C) = \infty$ using the lower bound from Hanneke et al. [2023]. We will make sure to point this out in the main text. With regards to $B(C)$ and $L(C)$, there can also be an arbitrary gap. Proposition 4 in Appendix F gives a class where $B(C) \leq 2$ but $L(C) = \infty$. We will make this explicit in the main text again.
> - Regarding the case when the rate is $O(1)$, Theorem 3 shows that when $B(C) < \infty$,  the minimax rate is at most $B(C)$ in the realizable setting. With regards to lower bounds, we can also show that when $T$ is large enough (namely $T >> 2^{B(C)}$), the lowerbound in the realizable setting is also $B(C)/2$. We will include this lower bound in the camera-ready version.
> - We thank the reviewer for pointing out these related previous works. We note that we do have a more in-depth discussion of prior work in Appendix A. In the final version, we will relocate this section to the main text. Additionally, we will incorporate a sentence that draws a precise comparison between our $B(C)$ and the rank notion from the paper by [Ben-David et al. 1997]. It is also important to note that while the dimension introduced by Devulapalli and Hanneke provides a lower bound, it is not feasible to establish an upper bound based on it. Specifically, their paper includes a theorem demonstrating the gap between transductive and self-directed online learning, even in the binary case.

---

> > ### Comment · Reviewer_Cx7F · 2024-08-12
> >
> > Thanks for these comments and the remark that in the realizable case the rate is $\Theta(B(C))$ if $T$ is large enough. I raised my score.

---

### Official Review · Reviewer_i2TD · 2024-07-16

**Soundness:** 3
**Presentation:** 3
**Contribution:** 2
**Rating:** 6
**Confidence:** 3

**Summary:**

This paper addresses the problem of multiclass transductive online learning with unbounded label spaces. The paper extends previous work on binary and finite label spaces to the more general case of unbounded label spaces. The authors introduce two new combinatorial dimensions - the Level-constrained Littlestone dimension and the Level-constrained Branching dimension - to characterize the optimal mistake bounds in this setting. They establish a trichotomy of possible minimax rates in the realizable setting, showing that the expected number of mistakes can only grow like \theta(T), \theta(log T), or \theta(1).

**Strengths:**

- The paper solves an open problem in online learning theory by characterizing optimal mistake bounds for unbounded label spaces.
- The paper is very well written and easy to understand.

**Weaknesses:**

- This paper extends the results of multi class transductive learning to infinite label space but the not clear how important is this setting.

**Questions:**

- Can the authors provide more more intuitive explanation of the Level-constrained Littlestone and Branching dimensions.
- Can the authors discuss the computational complexity of their algorithm for this setting?

**Limitations:**

Yes

---

> ### Author Rebuttal · Authors · 2024-08-05
>
> We thank the reviewer for noting that our work resolves an open problem in online learning theory and that our paper is very well written and easy to understand. We address each weakness and question below.
>
> - Multiclass learning with unbounded label spaces is a fundamental setting that has been under study for nearly 40 years, starting with [1,2] and more recently in [3,4,5,6]. Studying infinite label spaces is important as guarantees for multiclass learning should not inherently depend on the number of labels, even when it is finite. This is quite a practical concern as many modern machine learning paradigms have massive label space, such as in face recognition, next word prediction, and protein structure prediction, where the dependence of label size in learning bounds would be undesirable. Beyond being of practical interest, multiclass learning with infinite labels might also advance the understanding of real-valued regression problems [7]. Finally, in mathematics, concepts involving infinities often provide clearer insights into the true hardness of a problem. These motivations have been highlighted in lines 78-85.
> - To define the Level-constrained Littlestone dimension, we first need to define the  Level-constrained Littlestone tree. A Level-constrained Littlestone tree is a Littlestone tree with the additional requirement that the same instance has to label all the internal nodes across a given level. The  Level-constrained Littlestone dimension is just the largest natural number $d \in \mathbb{N}$, such that there exists a shattered  Level-constrained Littlestone tree of depth $d$. We will add a more intuitive explanation of this dimension in the final version.
> - To define the Level-constrained Branching dimension, we first need to define the  Level-constrained Branching tree.  The Level-constrained Branching tree is a Level-constrained Littlestone tree without the restriction that the labels on the two outgoing edges are distinct. The Level-constrained Branching dimension is then the smallest natural number $d \in \mathbb{N}$ that satisfies the following condition: for every shattered Level-constrained Branching tree $\mathcal{T}$, there exists a path down $\mathcal{T}$ such that the number of nodes on this path whose outgoing edges are labeled by different elements of $Y$  is at most $d$. We will add a more intuitive explanation of this dimension in Section 1.2 of the final version.
> - Our algorithms are not computationally efficient. Indeed, our algorithms require calculating the level-constrained Littlestone dimension or level-constrained branching dimension for concept classes. In the binary setting, the level-constrained Littlestone dimension equals the VC dimension, which is computationally hard to compute for general concept classes. That said, most online learning algorithms in online learning theory are not computationally efficient. For instance, in the case of adversarial online learning, SOA involves computing the Littlestone dimension of concept classes defined by the online learner in the course of its interaction with the adversary, which are challenging computations, even when the concept class and the set of features are finite [8]. Notably, no efficient algorithm can achieve finite mistake bounds for general Littlestone classes [9].
>
> [1] Balas K. Natarajan. Some results on learning. 1988.
>
> [2] B. K. Natarajan. On learning sets and functions. Machine Learning, 4:67–97, 1989.
>
> [3] A. Daniely, S. Sabato, S. Ben-David, and S. Shalev-Shwartz. Multiclass learnability and the ERM principle. 24th Conference on Learning Theory, 2011.
>
> [4] A. Daniely and S. Shalev-Shwartz. Optimal learners for multiclass problems. 27th Conference on Learning Theory, 2014.
>
> [5] N. Brukhim, D. Carmon, I. Dinur, S. Moran, and A. Yehudayoff. A characterization of multiclass learnability.  63rd Annual IEEE Symposium on Foundations of Computer Science, 2022.
>
> [6] S. Hanneke, S. Moran, V. Raman, U. Subedi, A. Tewari. Multiclass Online Learning and Uniform Convergence. 36th Conference on Learning Theory, 2023.
>
> [7] I. Attias, S.Hanneke, A.Kalavasis, A. Karbasi, G. Velegkas. Optimal learners for realizable regression: Pac learning and online learning. Advances in Neural Information Processing Systems 36 (2023).
>
> [8] P.Manurangsi, A. Rubinstein. Inapproximability of VC dimension and littlestone’s dimension. 30th Conference on Learning Theory, 2017.
>
> [9] A. Assos, I. Attias, Y. Dagan, C.Daskalakis,  M. K. Fishelson. Online learning and solving infinite games with an erm oracle. 36th Conference on Learning Theory, 2017.

---

### Decision · Program_Chairs · 2024-09-25

**Decision:**

Accept (spotlight)

**Comment:**

This is a theory paper on multiclass transductive online learning when the number of labels can be unbounded. All reviewers were very positive about the submission and found it to be a technically solid and valuable contribution to the field.